# Application of Silk-Fibroin-Based Hydrogels in Tissue Engineering

**DOI:** 10.3390/gels9050431

**Published:** 2023-05-22

**Authors:** Yihan Lyu, Yusheng Liu, Houzhe He, Hongmei Wang

**Affiliations:** Department of Pharmacology, School of Medicine, Southeast University, Nanjing 210009, China

**Keywords:** silk fibroin, hydrogel, scaffolds, cartilage, bone, skin, wounds, cornea, tympanic membrane, tooth, tissue engineering

## Abstract

Silk fibroin (SF) is an excellent protein-based biomaterial produced by the degumming and purification of silk from cocoons of the *Bombyx mori* through alkali or enzymatic treatments. SF exhibits excellent biological properties, such as mechanical properties, biocompatibility, biodegradability, bioabsorbability, low immunogenicity, and tunability, making it a versatile material widely applied in biological fields, particularly in tissue engineering. In tissue engineering, SF is often fabricated into hydrogel form, with the advantages of added materials. SF hydrogels have mostly been studied for their use in tissue regeneration by enhancing cell activity at the tissue defect site or counteracting tissue-damage-related factors. This review focuses on SF hydrogels, firstly summarizing the fabrication and properties of SF and SF hydrogels and then detailing the regenerative effects of SF hydrogels as scaffolds in cartilage, bone, skin, cornea, teeth, and eardrum in recent years.

## 1. Introduction

Silk fibroin (SF) has excellent physical properties, and its various secondary structures have different physical properties. The aggregated structure of SF can be roughly divided into crystalline and amorphous regions, with the regular, oriented α-helix, β-sheet, and cross-β-sheet structures in the former imparting high strength to SF [1]. SF also exhibits excellent biocompatibility and induces minimal or no inflammatory responses [2]. Moreover, thin films, hydrogels, sponges, nanoparticles, and other biomaterials can be made from SF due to its outstanding functional plasticity to meet different application requirements [3,4,5]. SF has been used in a variety of in vivo studies on cartilage, bone, skin tissue, etc.

Both natural and synthetic polymer materials are commonly used as raw materials for hydrogels. Natural materials, such as an extracellular matrix (ECM), are currently used more often. However, although scaffolds made of this material have excellent compatibility, they have problems such as high cost, poor mechanical properties, and significantly different raw material properties [6]. Synthetic polymer materials, such as PCL and PU, have solved the cost and source problems of raw materials; however, their tissue compatibility is poor due to their chemical properties [5].

SF-based hydrogels, made from SF as the raw material, with various designs, production methods, and combinations with other materials, exhibit excellent tissue repair functions that meet different clinical application requirements [7]. Widely used SF-based hydrogels not only have the advantages of a stable source and low cost but also have excellent advantages such as adjustable mechanical properties, strong combination with other materials, superior mechanical properties, biocompatibility, and controllable degradation and absorption [8,9]. For example, Tao et al. used the ultrasonication method to physically cross-link US-SF hydrogels and obtained better physical and chemical properties than previous materials. They also promoted cartilage regeneration by injection [10]. Some studies have used the simplest self-assembly method to mix SF with various materials to form composite hydrogels to maintain the biological activity of proteins and extend the degradation time, achieving better effects in bone healing [11]. In addition to bone and cartilage, SF gels have been used in skin wound repair. For example, Yu et al. used enzyme cross-linking to make SF/BG/THA hydrogels, which effectively promoted the activity of mouse fibroblasts and HUVECs, promoting skin regeneration [12]. There are also studies on SF-PPy hydrogels made with chemical cross-linking agents, which have excellent sensitivity, repeatability, and stability, to produce strain sensors to monitor various body movements. They can detect large and subtle movements of the human body [13]. SF composite hydrogels have also been applied to the cornea made by photo-cross-linking to improve mechanical stability and adhesion and reduce the degradation rate [14]. Finally, using the shear force method, SF and urea were mixed to make a hydrogel that, when printed with 3D technology, can promote the proliferation of human dental pulp stem cells [14]. Therefore, SF gels combined with other materials through different methods have a wide range of potential applications.

## 2. Preparation of SF Gels

### 2.1. Preparation of SF

The production of SF gel requires the raw materials of fibroin, which are typically obtained by a degumming process to remove sericin proteins from silk. Sericin functions as an adhesive to stabilize silk; however, it can cause inflammatory reactions and reduce biocompatibility compared with SF alone [15,16]. Furthermore, studies have shown that sericin-free fibers exhibit better tensile strain properties than sericin-containing fibers [17].

Currently, the most commonly used method for degumming is boiling under alkaline conditions. The cocoon is cut into thin slices and boiled in a sodium carbonate solution several times to obtain degummed silk. The degummed silk is then dissolved in a lithium bromide solution (9.3 mol/L) and subjected to a series of steps, including dialysis, recovery of high-concentration salt agents, filtration, weight measurement, and concentration determination, to obtain water-soluble SF or other products, which are stored at low temperatures (4–7 °C) [3,18]. Other high-concentration salts, such as calcium chloride or some alkaline organic solvents, can also be used to degum SF [19,20]. Additionally, different reagents, concentrations, ion types, pH values, heating times, and amounts can generate SF with different properties [18,21].

After obtaining degummed SF, gel production can proceed. SF-based hydrogel products can be broadly classified into two types based on their basic formation and cross-linking mechanisms: physical and chemical cross-linking. The difference between the two types lies in whether the SF molecules are bonded by covalent bonds between molecular chains.

### 2.2. Preparation of SF Gels

Various methods are used to prepare SF gels from the obtained degummed SF (Figure 1).

#### 2.2.1. Physical Cross-Linking

Physical cross-linking refers to methods that do not use chemical cross-linking agents. The fibroin molecules undergo a transformation from a random coil structure to a more unstable β-sheet conformation by physical processes such as self-assembly, shear force, ultrasonication, electric field, etc. Subsequently, the fibroin molecules in this low-energy state further aggregate through various non-covalent bonds, including hydrogen bonds and hydrophobic, electrostatic, and ion interactions, etc., ultimately forming a three-dimensional water gel structure.

Self-assembly

In the self-assembly method, random coil SF molecules in a regenerating solution are in a disordered state at high temperatures but can be induced to form a low-energy β-sheet conformation by evaporation or concentration at different temperatures, followed by self-aggregation to form a hydrogel. The gelation time of the RSF solution decreases as the temperature increases within a specific range due to increased molecular collisions [22,23].

Shear force method

The shear force method involves applying a sufficient shear force to the RSF solution to promote molecular interactions and induce gelation through concentration fluctuations [24,25]. For example, Tuna et al. reported that high-speed vortex shear treatment of SF solutions could effectively shorten the gelation time [25].

Ultrasonication

Ultrasonication is a faster and more effective physical strategy for preparing SF hydrogels by using ultrasound to generate pressure and temperature changes in localized areas of the hydrogel solution, thereby accelerating molecular interactions and inducing structural changes in SF to form a hydrogel. Ultrasonic power output, duration, and SF concentration are important factors influencing the state of the hydrogel [26,27].

Electric fields

Electric fields can also induce SF solution gelation; however, in this case, the random coil conformation transitions to an α-helix rather than a β-sheet conformation, allowing the transition process to be reversed with a change in external electric field polarity [28,29]. The electric-field-induced SF hydrogel formation process involves an influx of protons towards the positive electrode, causing the local pH to be lower than the isoelectric point of silk fibroin. This leads to the aggregation of negatively charged silk molecules near the anode, forming micelle-like structures, which then interact through the physical entanglement of molecular chains to form an SF hydrogel network [29,30].

#### 2.2.2. Chemical Cross-Linking

During the chemical cross-linking process, SF molecules are linked through covalent bonds using various catalytic methods, resulting in a spatially networked structure of water-soluble hydrogels. Due to the presence of covalent bonds, chemically cross-linked hydrogels generally exhibit superior physical stability and strength compared with physically cross-linked hydrogels. Currently, commonly used chemical cross-linking methods include chemical cross-linking agents, photopolymerization, γ-ray irradiation, and enzyme cross-linking.

Chemical cross-linking agents

Chemical cross-linking agents are added to the SF solution. Then, they react chemically with active functional groups in silk protein molecules such as -OH, -NH_2_, and -COOH to form a three-dimensional network structure of hydrogels. Commonly used chemical cross-linking agents include glutaraldehyde, genipin, EDC or EDC/NHS, and homobifunctional cross-linkers [31]. However, researchers have found that the hydrogels produced using chemical cross-linking agents may be toxic to tissue cells. Thus, reducing the use of chemical cross-linking agents in preparing SF gels is recommended [32].

Photopolymerization

During the photopolymerization process, a photoinitiator is added to an SF solution. Then, the SF solution is exposed to visible light or UV light of specific wavelengths to induce the absorption of sufficient energy by the initiator molecule to enter an excited state. Subsequently, it decomposes to generate free radicals, which initiate the polymerization of monomers and cross-linking to form covalent bonds, finally forming an SF gel. Rumeysa et al. reported that the adhesion properties of the material were affected by different exposure times under the same light exposure, necessitating strict control of factors such as the wavelength of light and exposure time when mixing with other materials and determining optimal conditions experimentally to reduce potential cytotoxicity [14,33].

γ-ray irradiation

γ-rays, due to their high penetrating power, can generate a large number of free radicals in the SF solution upon irradiation, causing an unstable chemical environment in the solution. After the recombination of free radicals, new covalent bonds are formed, followed by the rearrangement of molecular chains to form hydrogels [32,34]. It has been found that the polymer solution can be rapidly cross-linked and solidified in a very short time when preparing hydrogels using gamma-ray irradiation. Therefore, applying gamma-ray irradiation to produce SF gels is easy to operate and cost-effective and can prepare composite hydrogels mixed with other materials with uniform distribution and outstanding functionality [35,36].

Enzyme cross-linking

Unlike general chemical cross-linking, enzyme cross-linking does not generate cross-linking chemical bonds by reacting with active functional groups in SF or by exciting solutions to produce free radicals. Enzyme cross-linking mainly utilizes the unique catalytic activity of biological enzymes to catalyze specific groups on SF side chains to produce a three-dimensional network structure of cross-linking. Horseradish peroxidase (HRP) is the most commonly used enzyme cross-linking agent [37,38]. Therefore, enzyme cross-linking is more of a biological catalytic reaction, with simple and mild reaction conditions that avoid introducing chemical cross-linking agents, organic solvents, or other toxic substances into the product and have good biocompatibility. Therefore, it has become a commonly used research and application method [39].

Compared with physical cross-linking methods, chemically cross-linked SF gels exhibit more stable physical properties. Chemical cross-linking also allows for convenient and precise adjustment of various properties of hydrogels. Compared with photopolymerization and chemical cross-linking agents, enzyme cross-linking and radiation cross-linking methods are relatively safe and are currently the most commonly used chemical cross-linking methods. To prepare SF hydrogel materials with superior performance, multiple cross-linking methods are often used in practical applications, combined with other materials to produce superior functional composite hydrogels widely used in tissue repair and treatment (Figure 2).

## 3. SF Hydrogels for Cartilage

Trauma, inflammation, aging, and genetic abnormalities often lead to cartilage defects that impair joint function and even cause disability, posing a common medical challenge [40,41,42]. Cartilage lacks proliferative capacity since it cannot obtain rich nutrients or circulating progenitor cells. Hence, damaged cartilage is not often replaced by functional tissue and requires surgical treatment [5,43]. Early cartilage defects are partial-thickness cartilage defects (PCD) affecting the surface layer of cartilage, where the ECM (extracellular matrix) is missing and chondrocyte metabolism is hindered [41]. The ECM consists of collagen and proteoglycans and is responsible for the mechanical properties of cartilage tissue. Thus, maintaining and preserving the ECM is crucial for treating cartilage defects [43,44]. The drugs used to treat osteoarthritis (OA) in the early stage cannot cure it completely, and transplantation and stimulation methods are used in the late stage. Transplantation methods include allografts or autogenous cartilage transplantation, perichondrium and periosteum transplantation, osteochondral transplantation, and other graft repair techniques. Stimulation methods include joint cleansing and debridement, cartilage grinding and shaping, microfracture, joint traction, etc. Invasive total joint replacement is used to treat severe OA [45]. The current surgical transplantation treatment has its limitations. For example, autologous transplantation has a limited repair range and does not prevent the development of OA, and allogeneic transplantation is limited by the donor, which may lead to low graft fusion and disease transmission [46]. Currently, an injectable gel is the main development direction of cartilage tissue engineering (TE). Compared with open surgical implantation, the injectable gel can be minimally invasive, with fewer complications, better plasticity, and better physical and chemical properties, and it can encapsulate cells. They must be non-toxic and have a controllable gelling rate [45]. In addition, TE cartilage must have similar characteristics to native cartilage; the construction of osteochondral junction is also a challenge, and the mechanism of cartilage formation and regulation still needs to be studied. In addition, the affected articular surface of OA is larger, and the articular homeostasis is altered, which may enhance implant degradation. OA is often associated with obesity and age, and implant therapy is also affected [46] (Table 1).

### 3.1. Treatment of Osteoarthritis by Embedding Drugs

Wang et al. ground SF-BDDE hydrogel into pellets. Its advantage is that it can be used as a ball-bearing lubricant and stay in the joint cavity for over a month, effectively alleviating OA pain. After dissolving SF in lithium bromide (LiBr), SF-BDDE hydrogel was produced by cross-linking with butanediol diglycidyl ether (BDDE) and followed by freeze-drying. The SF-BDDE hydrogel was then ground into particles, and SF-BDDE gel spheres were prepared using an oil/water emulsification method [47]. In vitro, the metabolic activity of rat mesenchymal stem cells (rMSCs) cultured on the surface of the SF-BDDE hydrogel was enhanced, and the hydrophilicity of hydrogel was more conducive to cell distribution, adhesion, and proliferation. In vivo, SF-BDDE hydrogel spheres injected into the joints of osteoarthritic rats served as a carrier for biological lubrication and drug transport. Hydrogel spheres are not easily degraded by proteases; therefore, the release of drugs continues, effectively alleviating pain [47]. 

Similar to the role of SF-BDDE hydrogel, injectable SF-HA hydrogel can also serve as a carrier for sustained drug release. Hyaluronic acid (HA) is conducive to the survival of chondrocytes and the formation of an ECM. SF-HA gel can be cross-linked by forming Schiff bases between SF and HA, and the resulting HA is more stable and adhesive [48]. Methylprednisolone (MP), which is easily digested in the joint cavity, can be embedded in SF-HA hydrogel. The resulting MP-containing SF-HA gel can control the sustained release of MP, reduce inflammation for a long time, and create a good environment for the regeneration of chondrocytes [48]. In addition to MP, Song et al. improved the efficacy of bone marrow stimulation by embedding Etanercept (Ept), which is a clinically used anti-rheumatic drug. It can be embedded in SF hydrogel, achieving the effects of regulating cell metabolism and promoting cartilage regeneration [49]. Pullulan (PL) is a natural polymer derived from black yeast, which can promote the adhesion and differentiation of bone marrow mesenchymal stem cells (BMSCs). Song et al. coupled tylamine to carboxymethyl PL and then produced an Ept-containing SF-PL hydrogel by enzymatic cross-linking [39,49,50]. The Ept-containing SF-PL hydrogel has shown chondrogenic effects in both in vitro and in vivo experiments.

### 3.2. Materials Adapting to the Regenerative Environment

The weak adhesion of hydrogels to wet tissues is a challenge that should be overcome. Zhang et al. developed a multi-covalent bond AD/CS/RSF/EXO gel for this purpose [51]. Regenerated silk fibroin (RSF) is obtained by dissolving SF in LiBr [52]. The facilitating effect of chondroitin sulfate (CS) on cartilage is well-established. The alginate-dopamine (AD) conjugate is formed by covalently linking sodium alginate, and dopamine and is believed to mediate adhesion [51,53]. After combining AD, CS, and RSF, their degradation time is prolonged, and the activity of chondrocytes is well-regulated [54,55]. Exosomes (EXOs), extracellular vesicles secreted by BMSCs via paracrine mechanisms, can recruit BMSCs to the hydrogel and new cartilage through chemokine signaling pathways, promoting their migration, proliferation, and differentiation [56,57]. The AD, CS, and RSF solutions are concentrated, mixed, and embedded with exosomes, and the resulting AD/CS/RSF/EXO hydrogel is prepared by adding H_2_O_2_, and its high bonding strength is verified on the wet surface of porcine skin in vitro. In vivo, rats injected with the hydrogel showed cartilage defect regeneration and cell reshaping in the patellar groove [51]. In TE cartilage, incorporating exosomes aims to recruit chondrocytes to improve the effect of rapid gel degradation on ECM production. It has been demonstrated that it may achieve a better chondrogenic effect through hypoxic preconditioning [56,58]. BMSCs naturally exist in a hypoxic environment, leading to the upregulation of miR-205-5p expression in exosomes under hypoxia, which promotes chondrogenesis through the miR-205-5p/PTEN/AKT pathway [56,58].

Researchers often add cells or cytokines to improve the integration of hydrogels into the surrounding cartilage [45]. Transforming growth factor-β1 (TGF-β1) is a multifunctional growth factor that promotes bone and blood vessel formation and cell proliferation and can control the production of the ECM by regulating the synthesis of multiple proteins [59,60]. Blended injectable TGF-β1@SF/PCS/GP hydrogel was fabricated by Zheng et al. for cartilage regeneration [42]. Polylysine has high antibacterial activity, solubility, and thermal stability and good cell compatibility [61]. Chitosan (CTS) is a polysaccharide derived from crustaceans and obtained by deacetylation of chitin under alkaline conditions. Polylysine-modified chitosan polymer (PSC) is formed when CTS is functionalized with polylysine, resulting in enhanced the physical properties and antibacterial activity of CTS [42,61]. β-glycerophosphate (β-GP) serves as a catalyst to induce homogeneous gelation of CTS solution upon heating at physiological pH and temperature [62]. Moreover, incorporating GP provides the blended hydrogel with a more stable structure, crystallization properties, and better elasticity, viscosity, and degradability [42]. To make hydrogel, first, chitosan was dissolved in acetic acid, dissolved in polylysine, and neutralized by NaOH. Second, Polylysine-CTS was dissolved in HCl and mixed with RSF solution to make SF/polylysine-CTS solution. Then, GP was added to the solution to obtain SF/PCS hydrogel, and next, TGF-β1 suspension was dropped into the gel and lyophilized. Finally, BMSCs and human fibroblast cells were cultured on TGF-β1@SF/PCS/GP hydrogel and injected into the cartilage defect of rats for in vivo experiments [42]. In vitro, the blended gel enhanced the adhesion and proliferation of BMSCs, promoted the production of the ECM, up-regulated cartilage-specific gene expression, and down-regulated the expression of inflammatory factors. Moreover, there was no immune rejection in vivo [42].

### 3.3. Physically Cross-Linked Hydrogel Used for Chondrogenesis

However, it has been demonstrated that chemically cross-linked gels may cause cytotoxicity. For example, harmful free radicals are produced in the thiolene reaction, and residual aldehydes are produced in the Schiff base reaction. In addition, controlling the gelation process, the integrity of the gel, and the appropriate porosity are challenges that should be addressed for chemically cross-linked injectable hydrogels. In contrast, physically cross-linked gels are less toxic and easier to produce, although their mechanical properties are not as good as those of most chemically cross-linked gels. Moreover, physically cross-linked gels require high solid content, which will increase the cost, its viscosity, and patient discomfort [63].

Ultrasonic treatment of SF can rapidly form β-sheets, resulting in stable cross-linked SF gels [64]. Yuan et al. produced an injectable US-SF gel of embedded chondrocytes induced by ultrasound, demonstrating its chondrogenic ability in vivo [10]. Additionally, The SF(S)-COL gel prepared by Long et al. was sonicated: by mixing ultrasonically treated SF solution with collagen (COL) solution and injecting it into polydimethylsiloxane (PDMS), SF(S)-COL hydrogel can be produced [65]. In vitro, SF(S)-COL gel can promote the early expression of chondrogenesis-related genes and the deposition of sulfated glycosaminoglycans (sGAG). In an in vivo experiment using rabbits with joint injuries, this gel showed a longer degradation time and more regenerative potential [65]. The TGF-β1@CTS/BMP-2/SF gel prepared by LI et al. also used ultrasound-induced cross-linking: TGF-β1 promoted cartilage formation, and chitosan nanoparticles (NPs) were used as drug carriers for TGF-β1 [66]. TGF-β1@CTS gel can promote chondrogenesis. Bone morphogenetic protein-2 (BMP-2) can also work synergistically with TGF-β1 to promote chondrogenesis [59,67]. Therefore, by mixing TGF-β1@CTS, BMP-2, and SF solution and using ultrasonic treatment, TGF-β1@CTS/BMP-2/SF gel can be produced, which has been demonstrated to promote the chondrogenic differentiation of BMSCs in vitro and in vivo [59]. Basiri et al. also demonstrated that SF-DEWJ gel might be suitable for cartilage repair [68,69]. Wharton’s jelly (WJ) is a gelatinous connective tissue made up of the umbilical cord. Its ECM is similar to that of chondrocytes and contains a large amount of collagen, glycosaminoglycans, and various GFs [69,70]. Herein, decellularized extract from Wharton’s jelly (DEWJ) was obtained by sterilizing, slicing, and centrifuging the umbilical cords of full-term pregnant mothers. DEWJ was then mixed with ultrasonically treated SF solution and freeze-dried to obtain SF-DEWJ gel [68]. Human endometrial mesenchymal stem cells (hEnSCs) were cultured in SF-DEWJ gel and showed good survival and proliferation [68]. Moreover, experiments have shown that Wharton’s jelly ECM scaffolds are more conducive to chondrocyte proliferation and ECM deposition than articular cartilage ECM scaffolds [69], suggesting that SF-DEWJ gel may have cartilage regeneration potential.

In addition to widely used ultrasound cross-linking, Lee et al. used freeze–thaw cycle-induced physical cross-linking to prepare SF-PVA gel. [71]. The synthesized hydrophilic polymer polyvinyl alcohol (PVA) gel, made by freeze–thaw technology, has mechanical properties similar to natural cartilage, low production cost, and non-toxicity [72,73]. The freeze–thaw cycle technology involves repeated cycles of freezing and thawing. This physical cross-linking method relies on the pendant hydroxyl group structure of PVA [72,73]. The construction of SF-PVA gel aims to transform auricular cartilage by creating 3D auricular cartilage biomaterials, which can be applied in external ear reconstruction. Obviously, as a biomaterial that acts as ear cartilage, it should have physical characteristics similar to those of human ears, such as flexibility, softness, elasticity, and stability. SF-PVA gel meets these requirements [71,74]. Incorporating PVA enhances the tensile strength of the hydrogel and increases the gel’s pore size and thermal stability [71]. The production of SF-PVA gel involves dissolving PVA and SF solutions in water, using a salt immersion method, performing a freeze–thaw cycle, and then removing the salt to obtain the gel. To produce 3D auricular SF-PVA gel, PLA filaments are first 3D-printed to prepare the ear model, followed by a negative mold using silicone. Then, PVA and SF solutions are incorporated with salt particles into the silicone mold, and the subsequent freeze–thaw and salt removal processes are completed in the mold [71]. In vitro, cartilage cells cultured on this gel showed increased growth, metabolic activity, and the formation of lacunar cartilage structure and new cartilage. When the 3D ear-shaped gel seeded with cartilage cells was implanted into rats, mature ear-shaped engineered cartilage was obtained after six weeks of cultivation with no immunological or rejection reactions, indicating its great potential in auricular cartilage engineering [71].

Lee et al. fabricated GG/SF/CS gel by combining physical and chemical cross-linking methods to improve the biological and mechanical properties of hydrogels [75]. Chondroitin sulfate (CS), a sulfated glycosaminoglycan, is an important constituent of the cartilage ECM, which creates a quasi-natural microenvironment and adsorbs growth factors (GFs) to promote chondrocyte differentiation and migration, and ultimately alleviates pain and stimulates cartilage regeneration, although with limited mechanical properties [76,77,78,79]. Gellan gum (GG), a linear polysaccharide, becomes coil-shaped at high temperatures and then gelatinizes into a double helix structure at low temperatures. Incorporating GG hydrogel improves the mechanical properties of the entire composite hydrogel and synergizes with CS to create a quasi-natural microenvironment [75,80]. Composite hydrogel preparation involves first preparing a GG solution that is ionically cross-linked with CaCl_2_ (physical cross-linking), followed by uniform dissolution in CS. Then, an SF solution is chemically cross-linked with 1-ethyl-3-carbodiimide hydrochloride (EDC) before being incorporated into the GG/CS solution. Finally, the mixture is allowed to form a hydrogel which is embedded with rabbit chondrocytes for in vitro culture [75]. In vitro, the ECM exhibits good deposition, the encapsulated cells show good proliferation, differentiation, and migration, and cartilage-specific genes are upregulated [75].

### 3.4. Composite Scaffold Materials Based on Hydrogels

The SF hydrogel can also be combined with other materials to form composite scaffolds to improve mechanical properties, such as the SF-silk gel composite formed by combining SF hydrogel with silk microfibers [44]. The silk microfiber can be obtained by drying and degumming silk fibers in NaOH solution to enhance the mechanical properties of the hydrogel. The fiber–gel composite scaffold is prepared by mixing the silk microfiber suspended in PBS with primary bovine chondrocytes into the ultrasound-treated SF solution, which can promote the formation of cartilage fiber structure better than the single SF gel in vitro [44]. On the other hand, the CMCS-SF gel cross-linked by carboxymethyl chitosan (CMCS) and SF can also be made into a biomimetic fiber composite hydrogel scaffold PNFs/CMCS-SF gel by adding poly(3-hydroxybutyrate-co-3-hydroxyvalerate) (PHBV) nanofibers [81]. Chitosan is a natural polysaccharide with a structure similar to that of glycosaminoglycans, but it is insoluble in water [82]. However, CMCS, formed by chitosan’s reaction with monochloroacetic acid, is soluble in water and has excellent biological properties [83,84]. PHBV is a copolymer that can be made into a fiber scaffold by wet electrospinning, and it is added as a reinforcing agent to improve the mechanical properties of the SF gel [85,86]. To make the hydrogel, PHBV is first dissolved in chloroform to form a solution, and then wet electrospinning is used to obtain PHBV nanofibers. Next, the CMCS solution is mixed with the SF solution in equal volume; finally, the cross-linking agent poly (ethylene glycol) diglycidyl ether (PEGDE) and suspended wet-electrospun PHBV nanofibers are added. After cross-linking, freezing, and drying, the PHBV nanofiber-reinforced CMCS-SF gel (PNFs/CMCS-SF gel) can be produced [81]. Finally, the fiber–gel composite scaffold is added to the culture medium, and BMSCs are added for in vitro culturing, show cell proliferation, and ECM production under electron microscopy [81].

### 3.5. Three-Dimensional-Bioprinted Hydrogels Used for Chondrogenesis

The properties of TE cartilage are different from those of natural cartilage. Natural cartilage has a complex, layered structure, and the boundary between the layers is fuzzy, while the cells and the ECM between the layers are quite different. The mechanism of cartilage regeneration is also unknown. Therefore, it is difficult to reconstruct the bionic structure of traditional TE. Three-dimensional printing can assemble biomaterials into high-precision scaffolds, providing more possibilities for scaffold establishment and functional simulation [87]. However, because of the high requirements for the physicochemical properties of biological ink, such as viscosity, it is still necessary to determine the best manufacturing parameters.

Natural material gelatin (GT) is suitable for extrusion-based 3D printing due to its physical properties [88,89]. Therefore, by condensing gelatin with tyramine and dissolving it in SF solution, SF-GT hydrogel can be prepared by enzymatic cross-linking and low-temperature extrusion-based 3D printing. After seeding with stem cell polymers on the gel scaffold, SF-GT hydrogel with stem cell polymers promoted cartilage regeneration during in vivo experiments with rabbit cartilage defects [89]. In addition, the synthetic material polyethylene glycol (PEG) is also suitable for extrusion-based 3D printing due to its inertness, which provides better mechanical properties for SF/PRP/PEG hydrogel [90,91,92]. Due to the limitations of animal-derived growth factors (GFs) and recombinant GFs, platelet-rich plasma (PRP) can provide many autologous GFs, such as transforming growth factor-β (TGF-β), platelet-derived growth factor (PDGF), insulin-like growth factor-1, epidermal growth factor, and vascular endothelial growth factor (VEGF), which can promote cartilage cell proliferation [90,93,94,95]. PRP was extracted by centrifuging rabbit ear blood, and then freeze-dried SF was dissolved in the PRP solution and mixed with rabbit cartilage cells. After mixing with an equal volume of PEG solution, it could be used as a bio-ink to produce SF/PRP/PEG hydrogel by 3D printing [90]. Similar to drug release from gel, PRP, which originally had a short-term effect on GFs’ release, also achieved control and sustained release of GFs by combining with SF to form a hydrogel [93,96]. The in vitro culture of cartilage cells showed that SF/PRP/PEG gel increased the concentration of collagen and glycosaminoglycan and up-regulated the expression of cartilage-related genes [90]. It is well-established that poor integration of grafts with native tissues leads to cartilage degeneration and fibrocartilage formation. PRP is an autologous blood product that is relatively safe, easy to produce, and easy to integrate. However, the content of individual bioactive molecules in PRP is affected by the extraction process; therefore, the molecular composition of PRP and the appropriate dosage for cartilage regeneration should be explored further [97].

### 3.6. Hydrogels Suitable for OCD

Osteochondral defect (OCD) is the combination of cartilage damage and subchondral bone destruction after cartilage injury [98]. CuTA@SF gel, made by combining SF gel with metal–organic framework nano-enzymes, can improve OCD [99]. Tannic acid (TA), a natural polyphenolic compound with antioxidant and anti-inflammatory effects, can increase the stability of the hydrogel when cross-linked with SF [99,100]. Copper (Cu), as a trace element, also has antioxidant and anti-inflammatory properties [101]. When SF is cross-linked with TA to form a network, CuTA nano-enzymes composed of TA and Cu nanoparticles are dispersed therein, effectively improving oxidative stress and inflammatory reactions [99]. In vitro, CuTA@SF gel exhibits antibacterial and anti-inflammatory properties, enhances the osteogenic activity of BMSCs, and promotes ECM generation. In vivo, CuTA@SF gel promotes cartilage regeneration in rabbits with femoral OCD [99]. In addition, Zhang et al. developed an SF-LAP nanocomposite hydrogel using an enzyme cross-linking method, which can be used for the biphasic regeneration of cartilage and subchondral bone [37]. Studies have demonstrated that silicate nanoclay has the advantages of biocompatibility and a dual-charged surface and is widely used in various biological composite materials as a mechanical performance enhancer or bioactive agent [102,103]. Laponite (LAP) (Na^+^_0.7_[(Si_8_Mg_5.5_Li_0.3_)O_20_(OH)_4_]^−^_0.7_) is a disc-shaped silicate nanoclay that promotes bone and cartilage formation [104]. SF-LAP nanocomposite hydrogel can be prepared by dissolving synthesized LAP in ultrapure water, mixing it with SF solution, and adding horseradish peroxidase (HRP) and hydrogen peroxide (H_2_O_2_) [37]. According to the results, incorporating LAP accelerates the gelation of SF and enhances the mechanical properties and hydrophilicity of the gel [37]. Furthermore, when BMSCs are cultured in SF-LAP gel extract in vitro, their osteogenic and chondrogenic effects are enhanced. When the gel was implanted in the OCD rabbit model, synchronous regeneration of cartilage and subchondral bone was observed [37]. Subsequently, Zhang et al. also demonstrated that SF-MMT nanocomposite hydrogel, prepared using the same enzyme cross-linking method, could be used for biphasic regeneration [38]. Montmorillonite (MMT) [(Na,Ca)_0.33_(Al, Mg)_2_Si_4_O_10_(OH)_2_•*n*H_2_O], similar to LAP, is also a silicate nanoclay that has been used as a drug carrier and can be combined with SF and other materials to improve their physicochemical properties and biological activity [38]. This hydrogel has good mechanical properties similar to SF-LAP gel and can also promote the proliferation and differentiation of BMSCs in vitro, maintain the biological activity of chondrocytes, and promote the endogenous regeneration of cartilage and subchondral bone in vivo [37,38].

### 3.7. Challenges of Hydrogels in TE Cartilage

In recent years, TE cartilage has gradually approached natural cartilage in terms of performance and layered structure; however, it is currently insufficient for clinical applications. First, there are limitations in the fabrication methods and physicochemical properties; therefore, the material formulation and cellular microenvironment of the hydrogel should be explored. Second, the supply and proliferation technology of cartilage seed cells is limited. Third, the signal mechanism of cartilage regeneration and TE stem cell technology should be studied and developed to control stem cell transformation into chondrocytes in the gel. In addition, more clinical trials are necessary to explore whether the mechanical properties of hydrogels can withstand long-term loading. Finally, most approved clinical treatments are based on autologous chondrocytes, which can cause secondary damage to patients. The clinical translation of TE cartilage also faces issues such as cost and ethical regulations [45,87].

**Table 1 gels-09-00431-t001:** A summary of different types of silk fibroin hydrogels used in cartilage tissue.

Material(s)	Experimental Mode	Synthesis Method	Main Function	Ref.
SF-BDDE hydrogel spheres	In vitro and vivo	Chemical cross-linking agents; oil/water (o/w) emulsification	As lubricants and drug carriers in OA.	[47]
SF-HA hydrogel	In vitro and vivo	Chemical cross-linking agents; Ultrasonication	Load and controlled release of methylprednisolone in cartilage injury	[48]
SF-PL hydrogel	In vitro and vivo	Enzyme cross-linking	Promote Bone marrow stimulation by embedding Ept; Promote BMSCs chondrogenesis.	[39,49]
GG/SF/CS hydrogel	In vitro and vivo	ionic crosslinking; Chemical cross-linking agents	Promote the growth of encapsulated cells; Promote articular cartilage repair	[75]
AD/CS/RSF/EXO hydrogel	In vitro and vivo	Enzyme cross-linking	Promote BMSCs activity; Promoting rat patellar cartilage repair	[51]
TGF-β1@SF/PCS/GP hydrogel	In vitro and vivo	Chemical cross-linking agents	Promote the activity of L929 cells and BMSCs; regulate cartilage-specific gene expressions; anti-inflammatory; promote cartilage regeneration	[42]
SF-GT hydrogel	In vitro and vivo	Enzyme cross-linking; 3D printing	Promote chondrogenic differentiation of stem cells; Promote rabbit articular cartilage regeneration	[89]
SF/PRP/PEG hydrogel	In vitro	Chemical cross-linking agents; 3D printing	Controlled slow-release GFs; It is beneficial to chondrocyte and ECM formation.	[90]
US-SF hydrogel	In vitro and vivo	Ultrasonication	Ultrasonic crosslinking enhanced physical and chemical properties of gel; Suitable for cartilage regeneration	[10]
SF(S)-COL hydrogel	In vitro and vivo	Ultrasonication	Promote cartilage-specific gene expressions; Promote articular cartilage regeneration	[65]
TGF-β1@CTS/BMP-2/SF hydrogel	In vitro and vivo	Ultrasonication	Promote BMSCs to form cartilage	[59]
SF-DEWJ hydrogel	In vitro	Ultrasonication	Promote the activity of hEnSCs and chondrocytes	[68,69]
SF-PVA hydrogel	In vitro and vivo	Freeze-thaw cycle	Suitable for external ear reconstruction	[71]
SF-silk hydrogel	In vitro	Ultrasonication	Promote chondrocytes activity	[44]
PNFs/CMCS-SF hydrogel	In vitro	Wet-electrospun; Chemical cross-linking agents	Promote BMSCs activity	[81]
CuTA@SF hydrogel	In vitro and vivo	Enzyme cross-linking	Promote cartilage regeneration in femur OCD rabbits	[99]
SF-LAP nanocomposite hydrogel	In vitro and vivo	Enzyme cross-linking	Promote BMSCs activity; Promote synchronous regeneration of cartilage and subchondral bone	[37]
SF-MMT nanocomposite hydrogel	In vitro and vivo	Enzyme cross-linking	Promote BMSCs activity; Promote synchronous regeneration of cartilage and subchondral bone	[38]

## 4. SF Hydrogels for Bone

The main components of bone are water and organic and inorganic constituents. The organic component constitutes approximately 35% of bone weight, comprising 95% type I collagen fibers and 5% amorphous matrix. The amorphous matrix is gelatinous and is a protein and polysaccharide complex, which includes components such as hyaluronic acid, proteoglycans, bone sialoprotein, osteopontin, osteonectin, and osteocalcin [105]. The inorganic component accounts for approximately 60% of the bone weight, and the major component is hydroxyapatite (HAP) [106]. Silk is a natural protein biopolymer, mainly composed of gelatinous sericin and a core of silk protein. SF has been extensively studied in tissue engineering in recent years due to its excellent biocompatibility, low inflammatory reactants, physiomechanical properties, and ease of processing [107,108,109,110,111]. SF can generally be easily prepared into hydrogels by suitable processes without additional cross-linking reagents and toxic chemicals [64,110]. Hydrogels have a highly porous three-dimensional structure that functions as an extracellular matrix (ECM) mimic, providing a suitable environment for cell growth. Injectable hydrogels with good biocompatibility can re-expand to the desired shape according to the defect [112]. Fibroin has weak osteogenic properties; therefore, researchers prepare it into simple or composite fibroin hydrogels and add other bioactive substances to tap into its potential to repair bone defects (Table 2).

Some growth factors, represented by bone morphogenetic protein 2 (BMP-2), can significantly promote osteogenesis and are a promising alternative to bone grafting [113,114,115,116,117]. However, growth factor injections are associated with problems such as limited transmission, short half-life, side effects (ectopic bone formation and excessive inflammation of adjacent soft tissue), and poor economy [118,119]. SF hydrogels have become an appropriate choice as growth factor delivery systems in recent years because of their non-toxicity, controllability, and easy preparation [120,121,122]. Several studies have demonstrated that mixing fibroin hydrogels with bioactive factors (such as VEGF and BMP-2) improves the efficiency and quality of repair after bone defects. Diab et al. demonstrated that a delivery system composed of electrospun polycaprolactone (PCL) nanofiber mesh tubing and SF hydrogel is an effective carrier of BMP-2 [123], which can be used for the functional repair of large bone defects and significantly improve the biomechanical properties of bone. The effect of silk concentration on bone regeneration in vivo is thought to mask the effect of differential BMP-2 release in vitro, which is consistent with previous findings [124] where researchers used ultrasound-induced fibroin hydrogels as carriers of vascular endothelial growth factor 165 (VEGF165) and BMP-2 to fill the floor of rabbits’ maxillary sinuses, with a sustained release of VEGF165 and BMP-2 from fibroin hydrogels, promoting the repair of irregular bone defects. Guziewicz et al. proposed to control the release process by modulating growth factor loading and gel biodegradation rates [125], which might further optimize the process of bone regeneration. In addition, Ma et al. prepared composite hydrogels by filling polyethylene glycol-silk hydrogels (PEG-silk hydrogels) with polylactic acid (PLA) granules as carriers of BMP-2 [11], demonstrating that incorporating polylactic acid enhanced the hydrophobicity of PEG-silk hydrogel, better maintained the biological activity of the protein, and prolonged the degradation time, further exerting the potential of BMP-2 to promote bone regeneration. Based on its high water content, adequate mechanical strength, and easy-to-control gel process [44,64,126,127], Ding et al. reported that the SF–hydrogel/scaffold complex had good cell-carrying properties and confirmed that the SF–hydrogel/scaffold carrying stem cells could repair small bone defects of multiple shapes more efficiently [128]. Osteogenic progenitor cells were pre-induced to achieve calcium deposition and local mineralization by fibroin hydrogel-encapsulated stem cells before implantation in vivo, which might further shorten the bone repair time.

Hydroxyapatite (HAP), the main bone component in vertebrates, can significantly promote osteocyte proliferation and differentiation. Zaharia et al. incorporated HAP into SF hydrogels to obtain organic–inorganic hybrids similar to skeletal structures [129] and proposed ideas for biomedicine. However, HAP has low dispersibility in the aqueous phase, and HAP nanoparticles have poor compatibility with SF and thus tend to aggregate in aqueous SF [130]. One solution is to prepare SF hydrogels containing HAP nanoparticles by surface modification of HAP using the hyaluronic-acid–dopamine (HA-DA) conjugation method, which can be used as tissue engineering scaffold materials for bone regeneration [131]. Wang et al. introduced a novel coralline hydroxyapatite (CHA)/SF/glycol chitosan (GCS)/difunctionalized polyethylene glycol (DF-PEG) self-healing hydrogel that was combined with exosomes, effectively promoting bone healing in SD rat models by enhancing BMP-2 deposition and bone collagen deposition and maturation [132]. In addition, the incorporation of nano-hydroxyapatite (nHA)-graphene oxide (GO) mixed nanofillers with BMP-2 into fibroin hydrogels chemically prepared based on dopamine adhesives has also been promising in a novel bone regeneration therapy [133]. Kim et al. used γ-irradiation to convert HAP-dispersed SF solutions into chemically cross-linked three-dimensional porous SF composite hydrogels containing HAP nanoparticles (NPs) [36]. They found no effect of HAP concentration on the pore size inside the hydrogel and that the maximum compressive strength of the composite hydrogel decreased with increasing HAP content compared with the pure SF hydrogel, possibly due to insufficient organic/inorganic interactions. In addition, SF/HAP composite hydrogels promoted cell proliferation and adhesion while enhancing osteogenic differentiation of human mesenchymal stem cells (hMSCs) in vitro, indicating that 3D porous SF/HAP composite hydrogels are promising as biomaterials for bone tissue engineering.

Laponite (LAP) is an inorganic nanomaterial that promotes the osteogenic differentiation and bone regeneration of mesenchymal stem cells [134,135,136]. Magnesium ions, silicic acid, and lithium ions as non-toxic degradation products of LAP can further promote cell adhesion and osteogenesis [137,138]. Roohaniesfahani et al. incorporated MSM-10 ceramic particles into SF hydrogels by ultrasonic cross-linking, demonstrating that magnesium, strontium, and silicon ions released from MSM-10 ceramic particles increased the gelling time and silk I: silk II ratios in the secondary protein structure of SF hydrogels while improving the in vitro osteogenic properties of hydrogels [139]. Sun et al. prepared RSF/LAP hydrogels for optimized bone repair [140]. According to previous studies [104], LAP in RSF hydrogels can either increase the number of β-sheet domains that act as hydrogel cross-linkers or limit their growth. They found that RSF/LAP hydrogels degrade more slowly when the LAP content is high; thus, the space in the bone defect area is maintained for a long time, facilitating massive cell ingrowth and new bone formation. In addition, LAP in RSF/LAP hydrogels can mediate activation of the AKT signaling pathway in BMSCs to some extent [141,142], in turn upregulating the expression of osteogenesis-related genes.

Cheng et al. used a horseradish peroxidase (HRP)/hydrogen peroxide (H_2_O_2_) cross-linking system and electrospinning technique to construct a new SiNPsNFs-reinforced hydrogel to improve the mechanical strength and osteogenic induction ability of SF hydrogels alone [143]. SF composite hydrogels were prepared by covalently cross-linking phenol groups in tyrosine residues in aqueous SF with HRP and H_2_O_2_ [144,145], and SiFs were distributed in silk fibroin nanofibers (SF-NFs) using an electrospinning technique, and then pulverized into short NFs with a homogenizer and mixed with hydrogel precursors. The incorporation of SiNPsNFs not only improved the mechanical properties and stability of SF hydrogels by infiltrating the cross-linking network [146,147,148,149] but also effectively solved the problem that directly mixing NPs with hydrogel precursors would form aggregated NPs slurries [150,151]. However, since they could not completely inhibit the aggregation of SiNPs, partially aggregated SiNPs in hydrogels could partially mimic the distribution pattern of mineral crystals within the ECM (extracellular matrix) [152]. They found that SF composite hydrogels at a concentration of 5% SiNPsNFs stimulated both early and late osteogenic differentiation of pre-osteogenic MC3T3-E1 cells, consistent with previous studies [146,147]. In addition, the amino acid sequence (Gly-Ser-Gly-Ala-Gly-Ala)^n^ in SF may also improve cell adhesion, spreading, and proliferation [153,154]. In conclusion, SiNPsNFs-reinforced hydrogels could synergistically promote the activation of osteogenic differentiation through the osteogenic induction ability of SiNPs and the bioactive amino acid sequence in SF, inducing the formation of mineralized or new bone and laying the foundation for bone regeneration and repair by SF composite hydrogel without using any exogenous MSCs (mesenchymal stem cells) or growth factors.

**Table 2 gels-09-00431-t002:** A summary of different types of silk fibroin hydrogels used in bone.

Material(s)	Experimental Mode	Synthesis Method	Main Function	Ref.
polycaprolactone (PCL), SF solution, BMP-2	In vivo	Electric field	As an effective carrier of BMP-2 for functional repair of large bone defects	[123]
SF solution, BMP-2, VEGF-165	In vitro and in vivo	Ultrasonication	Promote repair of irregular bone cavity defects	[124]
SF, PEG, PLA/PLGA, BMP-2	In vitro and in vivo	Self-assembly	Maintains protein bioactivity and prolongs degradation	[11]
SF, HA, DA	In vitro	Ultrasonication	Used as tissue engineering scaffold material for bone regeneration	[131]
CHA/SF/GCS/DF-PEG hydrogels	In vitro and in vivo	Self-assembly	Facilitates bone collagen deposition and maturation	[132]
SF, GO, nHA, BMP-2	In vitro and in vivo	Click-chemistry cross-linking	Repair of bone defects	[133]
HAP, SF	In vitro	γ-Ray irradiation	Enhanced osteogenic differentiation of hMSCs in vitro	[36]
SF, MSM-10	In vitro and in vivo	Ultrasonication	Improving in vitro osteogenic properties of hydrogels	[139]
RSF, LAP	In vitro and in vivo	Ultrasonication	Mediates activation of AKT signaling pathway in BMSCs	[140]
SF solution, HRP, H_2_O_2_, SiNPs	In vitro and in vivo	Enzyme cross-linking.	Bone regeneration without any exogenous MSCs or growth factors	[143]

## 5. SF Hydrogels for Skin and Wounds

Skin tissue is composed of hormone glands, hair follicles, and other appendages, which resist bacterial infection, prevent pathogenic invasion, and retain moisture. It is the first line of defense of the human body’s immune system. The epidermis and dermis are composed of keratinocytes and an ECM. Complete loss of the integrity of the skin due to wounds, diseases, etc., will cause severe damage to body fluids, immunity, metabolism, and even death [5,155]. Skin wounds heal in three stages: inflammation, proliferation and differentiation, and matrix formation and remodeling [156]. The main routes for wound repair are the rapid formation of local blood vessels, deposition of collagen, and re-epithelialization [157]. Since the ideal wound dressing material should have good mechanical properties, water-holding capacity, biocompatibility, degradability, and structural flexibility for oxygen and nutrient supply [158], and because synthetic biomaterials often lack good biocompatibility and biodegradability, skin wound treatment at present relies on natural materials with good physicochemical properties, such as autologous grafts, decellularized dermal matrix allografts, xenografts, and naturally derived macromolecular scaffolds. The matrix commonly used in skin engineering is composed of protein polymers such as collagen or fibroin gel due to their similarity to the natural skin ECM. However, they have poor mechanical properties and are costly [159]. In contrast, many researchers have used SF and SF blends with excellent physical and biological properties in skin tissue engineering to change the activity of keratinocytes and fibroblasts [1]. In addition, SF-based gels are inexpensive, easy to manufacture, and biocompatible. Due to their similarity to the ECM, these gels can be used as scaffolds to provide a relatively humid environment for the wound [158]. Moreover, SF hydrogels retain the advantages of SF, supporting the proliferation of primary human fibroblasts and the migration of keratinocytes and promoting the deposition and remodeling of collagen fibers. They can also be used to deliver drugs, reduce pain, and accelerate the development and regeneration of complex skin layers with new blood vessels [158,159] (Table 3).

### 5.1. Polymer Blending or Surface Modification to Enhance Applicability

Currently, the research focus of TE skin scaffolds is on obtaining the ideal material through polymer blending [160]. For example, a CTS/SF/LP hydrogel, which is physically cross-linked using SF, chitosan, and l-proline (LP), has been demonstrated to promote skin wound healing [158]. As mentioned earlier, CTS is a natural cationic polymer commonly used as an extracellular matrix (ECM), with good swelling ability and composed of N-acetyl glucosamine and d-glucosamine, providing active sites [161]. LP is an imino acid that facilitates collagen synthesis [162]. Embedding LP in the CTS/SF gel improves its thermal stability, surface morphology, antioxidant activity, water retention, and degradability. Glycerol, a natural polymer, is used as a physical cross-linking agent to produce hydrogels to provide flexibility and elasticity [163]. Therefore, mixing CTS, SF, and LP solutions, adding glycerol, and transferring them to a mold, followed by drying and cleaning, produces a CTS/SF/LP hydrogel [158]. Cultivating NIH 3T3 L1 fibroblasts in the gel promotes fibroblast proliferation and fibrocyte adhesion, assisting in skin wound healing [158].

The current research on TE skin scaffolds focuses on enhancing cell adhesion and hydrophilicity through surface modification [160]. Pretreating or modifying commonly used biomaterials may lead to better results. For example, polarized hydroxyapatite enables SF gel to achieve excellent 3D and pore structures [164]. Carboxymethyl chitosan (CMCS) was more hydrophilic than CTS, and thiolated hyaluronic acid (THA) exhibited better cell adhesion than HA [12,165].

SF-pHAP hydrogels promote wound healing [164]. Hydroxyapatite (HAP) [Ca_10_(PO_4_)_6_(OH)_2_] ceramic has been used in bone tissue engineering, and dense HAP ceramic is a biomedical electret [164]. The electrically polarized HAP (pHAP) activates osteoblasts, and the polarized dense HAP ceramic contributes to bone formation because it stores electrical energy [166,167]. During production, pHAP powder was made from calcium hydroxide and phosphoric acid through sintering, filtration, freeze-drying, grinding, and electric polarization. After that, pHAP powder was mixed with SF solution and then fixed, dehydrated, and lyophilized to prepare SF-pHAP hybrid hydrogel [164]. Finally, the hydrogel was injected into the full-layer skin wound of porcine. The results showed that the gel promoted fibroblast maturation and collagen fiber formation. This regeneration of epithelium and stroma is related to its ability to store charge [164].

SF and carboxymethyl chitosan (CMCS) can be physically cross-linked through hydrogen bonding to form SF-CMCS hydrogel, which promotes wound re-epithelialization [165]. CMCS is a chitosan derivative with antibacterial, moisturizing, and antioxidant properties. The hydrogel is prepared using an electro-gelation method: SF and CMCS are mixed and refrigerated, NaCl is added to the SF-CMCS solution, and a low-voltage direct current is applied using a graphite anode, an Ag/AgCl reference electrode, and a platinum cathode. The gel is deposited on the anode, and the gel’s shape can be controlled by adjusting the deposition parameters. In vitro, the hydrogel exhibits water retention, stability, and antibacterial properties, and the gel extract promotes the proliferation of human embryonic kidney (HEK-293) cells. In vivo, it promotes granulation tissue formation and wound closure in full-thickness skin lesions in mice [165].

Terpolymer injectable SF/BG/THA hybrid hydrogel is prepared from thiolated hyaluronic acid (THA), SF, and bioactive glass nanoparticles (BG NPs). Its good mechanical properties and stability are attributed to its dual network structure [12]. HA exists naturally in the ECM and is an important component of glycosaminoglycan. It can mediate cell migration and ECM formation and is often used to produce hydrogels [168,169]. Compared with HA, THA relies on its disulfide bond and easily forms a gel. THA hydrogel has better adhesion but poor mechanical performance and other disadvantages [12]. Including BG NPs in hydrogel can lead to the controlled release of Si ions, promoting the migration of fibroblasts and endothelial cells to the wound surface [170]. After incorporating HRP, H_2_O_2,_, and BG NPs into SF solution and mixing with THA solution neutralized by NaHCO_3_ solution, the hydrogel can be cross-linked at physiological pH and 37ºC. In vitro, L929 murine fibroblasts and human umbilical vein endothelial cells (HUVECs) inoculated on hydrogels showed proliferation and migration. In vivo, hydrogel dressing stimulated angiogenesis and re-epithelialization in a full-layer mouse model with skin lesions [12].

Another example is the injectable CBPEGCTS/SF/PRP gel of a modified polymer blend [171]. Platelet-rich plasma (PRP) and polyethylene glycol (PEG) were mentioned in the previous SF/PRP/PEG gel used for cartilage injuries. PEG has biologically inert properties, while PRP can be separated from rat plasma and contains various GFs that promote tissue regeneration. However, it has a short half-life and is easily degraded by tissue proteases. The physically excellent glycol chitosan (GCTS) is a commonly used biomaterial in tissue engineering [171]. 4-carboxybenzaldehyde (CB) has been used as a catalyst [172]. CB functionalized PEG (CBPEG) can be prepared by dissolving PEG and CB in tetrahydrofuran, adding dicyclohexylcarbodiimide, and finally adding them all to diethyl ether. CBPEGCTS-SF gel can be prepared by dissolving powder forms of GCTS, CBPEG, and SF in normal saline and then mixing and embedding the PRP solution in this intermediate product, resulting in the CBPEGCTS/SF/PRP gel [171]. This hydrogel has high stability and adjustable degradation properties and protects PRP for the sustained and controlled release of GFs. In vitro, it promotes the proliferation of human dermal fibroblasts (HDFs) and human umbilical cord mesenchymal stem cells (HUMSCs). In vivo, it promotes the repair of epithelial, vascular, and neural tissues in type 2 diabetic rat wounds [171].

### 5.2. SF Gel in Conjunction with Plant Extracts

Plant extracts such as curcumin (CUR), Aloe vera mucilage (AVM), glycyrrhizic acid (GA), berberine (BER), etc., are added to SF hydrogel as anti-inflammatory materials for skin wound repair.

SF-CUR e-gel synthesized by the electro-gelation method can be used as a wound dressing for the controlled release of curcumin [173]. CUR [(E, E)-1,7-bis(4-hydroxy-3-methoxyphenyl)-1, 6-Heptadiene-3,5-ione] diketone compound, derived from the rhizome of turmeric, regulates transcription factors, inflammatory factors, and enzymatic activities and can achieve antibacterial, anti-inflammatory, antioxidant, anti-tumor, and wound healing effects [173,174]. However, it is not suitable for intravenous injection due to its poor hydrophilicity and stability [175]. Embedding curcumin into SF gel can increase its stability, pharmacological activity, and protein adsorption [173]. To prepare the gel, curcumin is dissolved in anhydrous ethanol and mixed with SF solution in a glass beaker. An electrode is placed at the mouth of the beaker, and a direct current is applied to produce the gel. According to one study, after freezing and drying, in vitro experiments showed that this e-gel effectively controlled the release of curcumin and exhibited significant antibacterial activity against *Escherichia coli* and *Staphylococcus aureus*. Immortalized human keratinocyte (HaCaT) cells cultured in the gel exhibited good distribution and activity [173]. Therefore, SF-CUR e-gel can enhance the healing activity of skin wounds. In addition, curcumin nanoparticles (Cur NPs) can be obtained by dissolving curcumin in dichloromethane, followed by stirring, centrifugation, discarding the supernatant, and drying [174]. A composite SF/PVA/Cur NPs hydrogel film can be prepared by adding polyvinyl alcohol (PVA) and Cur NPs to the SF solution, followed by heating and quickly spraying the mixture onto a glass plate [174]. Fibroblasts from mouse embryos cultured on the gel showed promotion of adhesion and proliferation. In vivo experiments on full-thickness skin defects in mice showed that this scaffold, like SF-CUR e-gel, had antibacterial and anti-inflammatory effects and promoted angiogenesis while promoting wound healing [174].

Studies have shown that SF/Ag/AV composite hydrogel photo-cross-linked by SF, silver nanoparticles (Ag NPs), and Aloe vera (AV) can also have antibacterial and anti-inflammatory properties and promote vascular and epithelial regeneration [176]. AV stimulates fibroblast proliferation and migration, while Ag NPs are natural antimicrobial agents. Controlled release can be obtained by wrapping with SF gel [177,178]. When AgNO_3_ was dissolved in SF solution, and AV was added after stirring, SF/Ag/AV hydrogel could be cross-linked after 24 h of illumination. In vitro antibacterial experiments have shown its antibacterial properties. The hydrogel can also release more Ag NPs in response to an acidic environment to realize anti-infection. In addition, the proliferation and migration of L929 fibroblast cells were demonstrated during culture on the gel. In vivo, during skin wound treatment in rats, the gel promoted collagen deposition and wound closure and promoted the polarization of macrophages towards the M2 regeneration phenotype, reducing scar formation. However, SF/Ag/AV hydrogel has also been shown to have a defect in that accumulation of Ag NPs in vivo may lead to organ damage [176]. Moreover, the SF/AVM/PVP hydrogel, formed by temperature-induced self-assembly and polymerization of SF, Aloe vera mucilage extract, and polyvinylpyrrolidone (PVP), can also promote epithelial tissue repair and wound closure [179]. Aloe vera leaf extract has antibacterial, anti-inflammatory, preservative, analgesic, and other effects. AVM is the gelatinous part of its leaves, containing sugars, vitamins, various enzymes, and antioxidants, which have been demonstrated to promote wound healing [180,181]. PVP has good hydrophilicity and can be used as a binder and stabilizer for hydrogel synthesis. It not only accelerates gelation but also enhances the adhesive properties of hydrogels. It is worth mentioning that the SF used to make this hydrogel includes BmSF from *Bombyx mori* and AaSF from *Antheraea assamensis* [179]. To make the hydrogel, the Aloe vera leaves are first peeled and chopped, and the latex and long fibers are removed before being freeze-dried into powder and dissolved in Milli-Q water. Then, BmSF and AaSF are mixed in proportion, and AVM and PVP are added, followed by storage at 37 °C to enable gelation [179]. In vitro, human dermal fibroblasts (HDF) and HaCaT cells cultured with SF/AVM/PVP hydrogel showed induced migration and proliferation. The gel also exhibited a controlled release of aloe molecules. In addition, in a rabbit skin wound model, SF/AVM/PVP hydrogel promoted the formation of granulation tissue and new epithelium, promoted ECM remodeling, and also played an anti-inflammatory role in early wound repair [179].

Reportedly, the photo-cross-linked injectable SF/GA/Zn hybrid hydrogel composed of interpenetrating polymer networks (IPNs) can modulate immunity and accelerate diabetes (DM) wound healing [182]. Glycyrrhizic acid is a natural antioxidant derived from licorice rhizomes and is often used to treat chronic inflammation. Glycyrrhizic acid self-assembles into a hydrogel in water due to its amphiphilic structure. This self-assembled GA hydrogel is injectable and has good biological properties. However, it has poor mechanical properties, low stability, and cytotoxicity at high concentrations [182,183]. Zn^2+^, as a bioactive component in the human body, plays a catalytic role in assisting and synergistically regulating immune and oxidative stress [184,185]. Methacrylate SF (SFMA) is synthesized from glycidyl methacrylate (GMA) and SF. For IPNs, GA self-assembly is induced by Zn^2+^. SFMA, GA, and Zn^2+^ are rapidly mixed to form the first network, followed by strengthening treatment with photoinitiator lithium phenyl-2,4,6-trimethylbenzoylphosphinate (LAP) to chemically cross-link and form the second network. The IPNs formed by the two networks provide the synthesized SF/GA/Zn hybrid hydrogel with excellent mechanical properties [182]. In vitro, this gel is non-hemolytic, and when mouse macrophages and rat fibroblasts are cultured on the hybrid hydrogel, increased cell activity is observed. In addition, the gel enhances the antioxidant capacity of macrophages, possibly achieved by simultaneously clearing exogenous oxidants and stimulating endogenous antioxidants [182]. In vivo, when the hybrid hydrogel is applied to a diabetic skin lesion rat model, it promotes the formation of a new epidermis and granulation tissue that resist the infiltration of inflammatory cells. The gel’s promotion of fibroblast proliferation may be related to its regulation of the macrophage phenotype transition [182].

The SF solution, berberine, and melanin are mixed and heated at 37 °C to form the SF/BER/melanin composite hydrogel, which has been demonstrated to have antioxidant properties and can be used for diabetic wound healing [186]. Melanin has antioxidative properties, and SF solutions containing melanin have both antioxidative and conductive properties [40]. Berberine is a natural isoquinoline alkaloid extracted from plants such as *Coptis chinensis* and berberis and has anti-inflammatory, antibacterial, anticancer, antioxidant, and metabolic regulation effects [187]. When BER is embedded in SF hydrogel, it is controlled and slowly released [186]. In vitro, SF/Ber/melanin hydrogel promotes the migration of NIH 3T3 fibroblasts. In an in vivo rat model of type 1 diabetic skin lesion, this hydrogel enriched macrophages, significantly reduced cell oxidative damage, and promoted collagen, granulation tissue, and blood vessel formation, accelerating wound healing [186].

### 5.3. A Skin Dressing Used for Sunscreen

Unlike most SF-based hydrogel skin dressings intended to repair wounds, mSF/TA/ZnO composite hydrogel has been designed for use in cosmetics and sunscreen lotions [188] since UV penetrates deep into the cortex, leading to collagen decomposition, photoaging, mutation, cancer, etc. ZnO, a mineral particle with photocatalytic activity, can absorb or reflect UV. In addition, ZnO is stable and non-toxic and does not affect visible light; therefore, it can be used as an additive for sunscreen hydrogel [188,189,190]. Modified SF (mSF) is a derivative of SF-BGE obtained by modifying SF through butyl glycidyl ether (BGE). Tannic acid (TA) is often used as a hydrogel binder or hemostatic synthetic material due to its easy bonding; it has antibacterial and antioxidant effects. In this composite hydrogel, TA forms coordination bonds with ZnO and hydrogen bonds with mSF [188,191]. During the production process, SF was dissolved in CaCl_2_ and ethanol solution, BGE was added by drops, and mSF was obtained after high temperature, dialysis, centrifugation, and lyophilization. Then, TA solution was added to the mSF solution, and ZnO NPs were added to prepare mSF/TA/ZnO composite hydrogel. Finally, in vitro and in vivo experiments showed that this gel is not only UV resistant, hemostatic, antibacterial, and antioxidant, but it is also non-irritating to the skin under the condition of appropriate ZnO, which is applicable to sunscreen lotions [188].

### 5.4. Flexible Wearable Electronic Products

In addition to being used as a skin dressing to promote skin tissue regeneration, SF gel can also be used as a flexible wearable electronic product such as epidermal sensors and electronic skin [13,192,193]. These products can convert motion and physiological signals on the skin surface into electrical signals [194]. For example, flexible wearable microelectronic sensors can detect wounds and provide ideas for designing a new generation of TE skin dressings. Wound treatment strategies may be accurately and timely adjusted with the help of flexible wearable sensors. They can also be connected to materials with functions such as hemostasis and conduction, making their functions more diverse [160,195].

For example, SF-PPy hydrogel is manufactured for use in flexible epidermal sensors [13]. Previous studies have shown that adding conducting polymers, such as polypyrrole (PPy), to the gel can increase its conductivity [196]. Therefore, SF gel is soaked in pyrrole monomers and FeCl_3_ solution, and after chemical polymerization, a black SF-PPy hydrogel is obtained. Through resistance testing and other tests, this hydrogel has shown excellent conductivity, mediated by PPy in situ polymerization and network formation in the gel. In addition, SF-PPy hydrogel has good flexibility and is suitable for deformation. When the hydrogel was attached to the finger joints and neck of the subject with tape for activity detection, the results showed that it could sensitively identify motion amplitudes. Therefore, SF-PPy hydrogel, as an epidermal strain sensor, demonstrates good sensitivity, reproducibility, and stability [13]. Similarly, SF/PVA/TA/BX hydrogel is also suitable for flexible epidermal sensors [192]. Borax (BX) can decompose into an ion state in water, which provides the gel with conductivity [197]. Polyvinyl alcohol (PVA) is often used for cross-linking in gels due to its easy bonding. It can form a reversible borate ester bond with BX, making the gel self-healing. TA in the mSF/TA/ZnO hydrogel can enhance the adhesion and antibacterial ability of the hydrogel. Combined with SF, the gel has better self-healing and adjustable viscoelasticity. It is worth mentioning that incorporating SF into the gel enhances the mechanical properties but serves as a reinforcing agent to solve the non-Newtonian fluid behavior caused by adding PVA and BX to the gel [192]. The SF/PVA/TA/BX hydrogel is prepared by mixing PVA solution with SF solution and BX solution with TA solution. Characterization tests have shown that this hydrogel has good conductivity, stretchability, strain sensitivity, and stability, similar to SF-PPy hydrogel, in addition to self-healing and antibacterial properties. It has strong adhesion, easy detachment, and repeatability. It can detect joint movements and subtle movements such as swallowing and smiling. In addition, it can distinguish the amplitude of laughter and breath [192]. The transparent, strain-sensing SF/CNC/PAM hybrid hydrogel comprises SF, cellulose nanocrystals (CNC), and polyacrylamide (PAM). Its properties are similar to SF/PVA/TA/BX hydrogel, with excellent mechanical properties, adhesion, ionic conductivity, and self-healing ability due to its internal physical cross-linking network [193]. CNC, as an additive, increases the stability and elasticity of the gel, while PAM, as a water-soluble polymer, is commonly used in gel preparation due to its ease of polymerization to provide physical cross-linking and enhance the mechanical properties of the gel [193,198]. To obtain CNC powder, citric acid (C_6_H_8_O_7_) and HCl are mixed, and microcrystalline cellulose (MCC) is hydrolyzed at a high temperature, followed by washing with deionized water and freeze-drying. CNC powder is then added to the SF solution along with acrylamide, the cross-linking agent bromoacetic acid, the conductive material KCl, catalyst ammonium persulfate (APS), and tetramethyl ethylenediamine (TEMED). Finally, the SF/CNC/PAM hybrid hydrogel is polymerized under light (UV) [193]. In performance testing, this gel’s strain sensitivity is reflected by detecting changes in limbs, vocal cords, and pulses. It can also recognize ambient gas molecules such as formaldehyde and ammonia vapors. It can even serve as a temperature indicator after adding reversible thermochromic pigment (RTP) by changing color when an abnormal body temperature is detected [193].

There is also an SF-PAAm double network (DN) hydrogel adhesive manufactured for use under the epidermal sensor and insulation layer to enhance adhesion with the human skin [199]. Polyacrylamide (PAAm), which has biocompatibility and hydrophilicity, can interfere with hydrogen bonds in the SF gel to improve the water retention of the gel [199]. The SF-PAAm DN hydrogel adhesive has adjustable adhesion and can strongly and durably adhere to the wet skin surface. This DN hydrogel can be prepared by mixing SF, acrylamide, ammonium persulfate, and tetramethylethylenediamine (TEMED) and heating. In motion tests on subjects’ sweaty forearms, the DN hydrogel adhesive exhibited strong adhesion and was integrated into arrays of different epidermal sensors [199]. In addition, SF hydrogel and SF/PVA hybrid film can be combined to form electronic skin (e-skin) that functions as a capacitive pressure sensor [200]. Concerning the composition of this e-skin, one is the SF hydrogel cross-linked with acetate esterase, which is highly elastic and used as the pressure-sensitive layer and the dielectric layer of the sensor. On the other hand, the SF-PVA film is made by mixing SF, polyvinyl alcohol (PVA), and glycerin and pouring it onto a release paper for use as the sensor’s electrode. When using the SF-PVA film, a template is drawn on it with a graphite pencil, and a copper wire is attached to the side. Then, the hydrogel is sandwiched between the two films and fixed with resin adhesive to form an SF sensing unit, which can be expanded to become an e-skin. The test results of the e-skin show that it has good stability, biocompatibility, and biodegradability and excellent sensitivity and can quickly respond to small changes in human joints, vocal cords, and pulses [200].

### 5.5. Composites Based on SF Hydrogels

SF gel-related composite materials are also used in skin and wound tissue engineering. The BFGF-loaded liposome with an SF hydrogel core (SF-BFGF-LIP), made by combining SF gel, liposome (LIP), and basic fibroblast growth factor (BFGF), has been demonstrated to promote wound healing [157]. Exogenous growth factors promote wound healing by stimulating cell proliferation and the formation of blood vessels and ECMs. These growth factors include epidermal growth factor (EGF), vascular endothelial growth factor (VEGF), fibroblast growth factor (FGFs), platelet-derived growth factor (PDGF), etc. [201]. BFGF can promote the formation of fibroblasts and capillaries, but its disadvantages are that it has a short half-life and is easily degraded by proteases in the wound exudate [157,202,203]. Liposomes may contain hydrophilic proteins in their inner aqueous phase. Controlling the slow release of their contents leads to a moist environment on the skin surface, facilitating wound healing [204]. Although the direct embedding of BFGF in the aqueous phase of the LIP can increase its in vitro stability, it is also prone to leakage in the traumatic exudate [204]. Therefore, SF gel has been introduced as the core of the LIP with the reverse-phase evaporation method. The BFGF solution was mixed with SF solution, and the aqueous SF-BFGF solution was added to the oil phase composed of lipids, cholesterol, and methylene chloride, which was emulsified after ultrasonic treatment. Then, the residual dry lipid film was hydrated after steaming the emulsion. The SF gelation was induced by ultrasound again, and finally, the SF-BFGF-LIP was prepared [157]. This process makes BFGF more stable and prevents it from leaking out of the LIP, resulting in a sustained slow release of BFGF [157]. In vitro, NIH 3T3 fibroblasts were cultured in SF-BFGF-LIP; the wound exudate exhibited cell migration and proliferation. In vivo, SF-BFGF-LIP treatment of second-degree scald wounds in mice showed that it induced the proliferation of epithelial and dermal cells, promoted collagen deposition, and promoted the formation of blood vessels and granulation tissue, accelerating wound healing [157].

### 5.6. Challenges of Hydrogels in TE Skin

Currently, the development of anti-inflammatory hydrogel dressings is still in its early stages. First, considering the complexity of human wounds, which in animal models are often difficult to replicate, and hydrogel dressings may also induce immune reactions such as infection and allergy [195]. Second, different types of wounds have different degrees of injury and exudate, and it is difficult for scaffolds to meet all patient requirements; therefore, it is necessary to combine various materials to produce multifunctional composite scaffolds. Third, problems such as vasoconstriction, insufficient vascularization, and scar formation should be solved after wound healing. Finally, regulatory and ethical issues must be resolved in clinical translation [160].

From the perspective of flexible wearable electronic devices, their application is limited by their high Young’s modulus, poor biocompatibility, and poor responsiveness [205]. Moreover, although most current self-healing e-skins can achieve almost complete self-repair, their life cycle is short, leading to large amounts of waste [206]. More hydrogel electronic devices should be integrated to realize their multiple functions, such as sampling and information communication, and basic characteristics, such as flexibility, biocompatibility, and comfort. In addition, flexible hydrogel electronic devices should meet the needs of different situations [205]. In conclusion, flexible electronics is an emerging research field that depends on the development and cooperation of multi-disciplinary technology [205].

**Table 3 gels-09-00431-t003:** A summary of different types of silk fibroin hydrogels used in skin.

Material(s)	Experimental Mode	Synthesis Method	Main Function	Ref.
SF-pHAP hydrogel	In vitro and vivo	Electric field; Self-assembly	Promote fibroblast maturation; Promote wound healing in porcine	[164]
SF-CMCS hydrogel	In vitro and vivo	Electric field	Promote proliferation of HEK-293 cells; Promote wound closure	[165]
CTS/SF/LP hydrogel	In vitro	Physical crosslinking mediated by glycerol	Promote fibroblast activity;	[158]
SF/BG/THA hydrogel	In vitro and vivo	Enzyme cross-linking	Promote murine fibroblast and HUVECs activity; Promote skin regeneration	[12]
SF-CUR e-hydrogel	In vitro	Electric field	Promote human keratinocyte activity	[173]
SF/PVA/Cur NPs hydrogel film	In vitro and vivo	Self-assembly	Promote fibroblast activity; Antibacterial and anti-inflammatory; Promote skin regeneration	[174]
SF/Ag/AV hydrogel	In vitro and vivo	Photopolymerization	Controlled release of Ag NPs for antibacterial; Promote fibroblast activity and wound healing	[176]
SF/AVM/PVP hydrogel	In vitro and vivo	Self-assembly	Promote HDF and HaCaT activity; Anti-inflammatory and regenerative in skin damage	[179]
SF/GA/Zn hydrogel	In vitro and vivo	Photopolymerization	Promote macrophage and fibroblast activity; Promote diabetic wound healing	[182]
SF/BER/melanin hydrogel	In vitro and vivo	Self-assembly	Promote fibroblast activity; Promote diabetic wound healing	[186]
CBPEGCTS/SF/PRP hydrogel	In vitro and vivo	Chemical cross-linking agents; Self-assembly	Controlled release GFs; Promote HDF and HUMSCs activity; Promote diabetic wound repair.	[171]
mSF/TA/ZnO hydrogel	In vitro and vivo	Photopolymerization; Self-assembly	Sunscreen for UV protection	[188]
SF-PPy hydrogel	In vitro and vivo	Chemical cross-linking agents	Flexible skin strain sensor	[13]
SF/PVA/TA/BX hydrogel	In vitro and vivo	Electric field	Flexible skin strain sensor	[192]
SF/CNC/PAM hydrogel	In vitro and vivo	Photopolymerization	Flexible skin strain sensor	[193]
SF-PAAm DN hydrogel	In vitro and vivo	Self-assembly	Adhesive for flexible skin electronic equipment	[199]
SF hydrogel; SF/PVA film	In vitro and vivo	Solution casting method; Enzyme cross-linking	Electronic skin	[200]
SF-bFGF-LIP	In vitro and vivo	reverse-phase evaporation method; Ultrasonication	Promote fibroblast activity; Promote skin regeneration	[157]

## 6. Application of SF Hydrogel in the Cornea

The cornea is a transparent, avascular, highly innervated tissue located in the anterior wall of the eyeball and functions as a refractive and protective intraocular structure [207,208,209,210]. It is estimated that approximately 23 million people worldwide are currently blind due to corneal diseases [208,211]. Since the discovery of limbal stem cells (LSCs), autologous or allogeneic limbal tissue transplantation has been thought to reconstruct the injured ocular surface [212]. However, corneal transplantation, as the most important treatment for corneal diseases, is limited by donor shortages and rejection after transplantation [213,214,215]. Bioengineered corneas have attracted researchers’ attention because they may address critical issues such as tissue availability and rejection. Corneal regenerative materials must have water content and transmittance similar to the natural cornea and be stable to resist enzymatic hydrolysis, have good mechanical properties to withstand surgical sutures, and be biocompatible to support the growth of corneal epithelial cells [216]. In this context, the inherent optical clarity of fibroin, combined with controllable degradation rates and mechanical properties, as well as the ability to mimic multiple features of the native extracellular matrix (ECM), make it a promising candidate [217,218].

Researchers have tried to prepare SF hydrogels suitable for reconstructing corneas by different cross-linking methods in recent years. Genipin (GP) [219,220], as a natural cross-linking agent, can react with free amino groups in fibroin. Zhou et al. prepared porous polyvinyl alcohol/fibroin/nano-hydroxyapatite (PVA/SF/n-HA) composite hydrogels with good physical properties by GP cross-linking [221]. Chemical cross-linking with GP results in better tensile strength, thermal stability, and biocompatibility of sacrificial parts of PVA/SF/n-HA composite hydrogels. Human corneal fibroblasts (HCFs), which grow in the pores of composite hydrogels and exhibit good adhesion and proliferation properties, are a promising method for repairing damaged corneas. It has been shown that SF interpenetrating network hydrogels have high strength and toughness and have high application value in the biomedical field [222]. Bhattacharjee et al. used different ratios of SF relative to polyacrylamide (PA) to prepare semi-interpenetrating hydrogels [223], demonstrating that these semi-interpenetrating networks could promote keratinocyte migration during healing, improve cell adhesion, and improve the efficiency of in situ corneal tissue regeneration. The number of hydrophilic groups (-NH_2_/-CO) increases as the concentration of fibroin (wt%) increases, which in turn improves the permeability of oxygen, nutrients, and water-soluble metabolites to help form new tissue structures. In addition, hydrolysis of the amide group (-NH_2_) to the carboxyl group (-COOH) increases osmolality within the network, which might contribute to composite hydrogel swelling. Hydrogel pores shrink as SF content increases, but the number of pores and the pore walls increase, improving the overall toughness of the hydrogel [224]. Mixing SF solutions at concentrations of 10–15 mg/mL with various polar organic solvents (acetone, ethanol, methanol, and isopropanol) was reported to drive the assembly of silk micelles into submicron-sized particles (<100 nanometers) [225], consistent with a previous study in which SF precipitated in ketone libraries to form uniform silk nanoparticles [226]. Using this property, transparent nanostructured SF hydrogels were prepared by mixing acetone with SF solution at a ratio of 1:2 and evaporating acetone at room temperature while adding deionized water to the gel exchange [225]. Nanostructured fibroin hydrogels are biocompatible and allow the attachment and proliferation of human skin fibroblasts. In addition, seeding human corneal epithelial cells (HCECs) on the surface of hydrogels forms epithelial cells without changing the transparency of the gel.

Traditional corneal sutures have disadvantages such as infection and complicated operation. Tissue adhesives have become promising alternatives for suturing corneal incisions because of their simple application and less tissue damage [227,228]. Methylacryloyl gelatin (GelMA), although an ideal adhesive, is weakly adhesive to ocular surfaces [229,230,231]. To address this problem, Tutar et al. incorporated SF solution into the GelMA solution prepared by standard procedures, added the photoinitiator Irgacure 2959, and finally, applied UV irradiation to obtain adhesive composite hydrogels [14]. With its good light transmittance [232], incorporating SF improves the mechanical stability and adhesion of pure GelMA hydrogels and reduces the degradation rate. In addition, researchers found that the mechanical properties of adhesive composite hydrogels improved with increasing UV cross-linking time; however, increasing β-sheet in the structure resulted in increased hydrogel brittleness when the light-curing time reached 90 s, which was considered the optimal UV light-curing time based on multiple experimental results [14]. In addition, given that bacterial infections may cause corneal scarring, prophylactic antibiotic eyedrops are often the standard procedure after ocular surgery. However, rapid drainage and irrigation difficulties make antibiotic eyedrops less effective and place postoperative wound healing at risk [233,234]. Relying on its hydrophilicity to incorporate antibiotics and anti-inflammatory agents into SF provides new ideas to address this problem [235]. Glyceryl methacrylate (GMA) (Sigma-Aldrich) was incorporated into the degummed SF solution, and the lysine amino acids of silk were modified by methacrylate groups to make injectable and lightly polymerizable SilkMA solutions, and covalent cross-linking of SF hydrogels was achieved by photo-cross-linking by adding photoinitiators and UV light-curing [236]. The light transmittance results obtained for 15% and 20% SilkMA hydrogels matched the values reported in the literature for healthy human corneas (78–80%) [237]. Light-cured hydrogels improve mechanical strength and provide greater flexibility in controlling in situ cross-linking kinetics [208,211]. SilkMA hydrogels have swelling properties that are inversely proportional to polymer concentration, and moisture content results tend to be similar, with good in vitro degradability and the ability to support adhesion and proliferation of human dermal fibroblasts (HDFs) [238,239]. More importantly, gentamicin-loaded light-cured SilkMA hydrogels effectively inhibit the growth of *Staphylococcus aureus* and *Pseudomonas aeruginosa*, enhancing the potential clinical efficacy of this corneal substitute.

Similarly, researchers are trying to tap into the potential of SH to improve vision. Flavin mononucleotide (FMN), a biosensory blue light protein cofactor produced by riboflavin kinase-mediated riboflavin excitation, can induce coupling reactions between free radicals and serine protein tyrosine residues and form a dityrosine complex that binds SF together, thereby preparing transparent hydrogels with almost-pure elastic mechanical properties [240,241]. The advantage of this light-cured hydrogel is that it can be generated by visible light rather than potentially damaging ultraviolet radiation. In addition, photo-cross-linking of SF using riboflavin formed tight binding to native corneal collagen, in short, to indicate a layer of silk covering the cornea, making it possible to change the optical thickness of the cornea with a high spatial resolution to achieve almost safe vision correction [242,243].

Bioengineering materials other than SF for corneal repair and regeneration is continuously being explored and developed. Various polysaccharides such as cellulose, chitosan, alginate, agarose, and hyaluronic acid have been widely explored as scaffold biomaterials for tissue-engineered production of the cornea. However, clinical translation of polysaccharide-based corneal implants is challenging due to the rather demanding regulatory requirements for keratoprosthesis implants and because the final shape and quality of polysaccharide-based corneal implants can change due to differences between batches [244]. Different types of cells have been transplanted onto various scaffolds for tissue regeneration. Amniotic membrane sheets and extracts, fibrin glue, and viable autologous human corneal epithelial cells have been used as commercial products for treating corneal disorders. However, their widespread use worldwide is hampered by legal and religious restrictions, the lack of suitable donors, the risk of infection, and the need for advanced preparatory equipment [245,246,247,248]. Bioprinting technology enables layer-by-layer construction of composite structures, and 3D scaffolds using traditional tissue engineering methods have been widely designed in recent years [249]. However, current techniques to construct functional artificial tissues require higher spatial complexity, better cell–cell interactions, and vascularization. Traditional 3D printing methods cannot precisely control pore geometry and pore size or create regions of change within a single scaffold. In addition, the morphology of the construct may differ from the actual shape of the tissue due to the limited resolution of the scanner. Therefore, these challenges may lead to longer operative times and damage to surrounding tissues [250,251]. Organoids with organ-level function have advantages in ocular physiology, ocular disease models, drug screening, or gene therapy vectors; however, the application of organoids is also limited by many limitations, such as metabolic waste accumulation, reduced glucose/oxygen diffusion, low variability, and incomplete maturation. Over the past two decades, cell sheet methods have been developed to regenerate various tissues or organs. Functional organelle-like 3D structures with high tissue bionics can be fabricated by cell sheet methods. However, the prepared organs exhibit poor elasticity and rigidity due to the lack of scaffolds. In addition, the high cost of preparation is also a challenge that should be addressed clinically by the cell sheet method [252,253].

## 7. Application of SF Hydrogels in Teeth

Pulp tissue is located in the pulp cavity of teeth, is rich in blood vessels and neural networks, and is extremely sensitive to external stimuli, which are essential for tooth health [254,255]. Damaged pulp due to infection, among other causes, tends to make regeneration difficult [256]. Root canal therapy is indicated for irreversible pulpal lesions to remove the pulp; however, it increases the risk of residual dentin infection [257,258]. Since the anatomy of the root canal system is complex and irregular, an injectable system with low invasiveness and high sterility that can accommodate irregular voids is preferable to replace solid stents [259,260,261]. Injectable hydrogels mimicking the viscoelasticity of the native soft tissue extracellular matrix and carrying and delivering stem cells for pulp regeneration have emerged as promising alternative treatments [262] (Table 4).

Chen et al. developed an extrudable printed RSF-based presolidified hydrogel using high-molecular-weight RSF (HMWRSF) and carbamide as raw materials and demonstrated good cytocompatibility for dental pulp mesenchymal stem cells and the ability to support their growth and proliferation [263]. Similarly, fibroin hydrogels loaded with rapidly gelatinized cells have exhibited good compatibility with human dental pulp stem cells (hDPSCs) by visible light-induced cross-linking of riboflavin as a photoinitiator [264]. Stem cells from human exfoliated deciduous teeth can proliferate on silk sponges, but whether scaffolds can be used for root canal repair requires further in vivo studies to confirm [265].

Modified SF and hyaluronic acid (HA) link highly reactive vinyl groups to form an RSFMA/MeHA solution, incorporating Irgacure 2959 photoinitiator and exposing it to UV light to make a photo-cross-linked SF/HA (RSFMA/MeHA) hydrogel [266]. RSFMA/MeHA composite hydrogels at appropriate concentrations (1% MeHA/3% RSFMA) promoted odontogenic marker expression and mineralized matrix deposition in primary cultures of hDPSCs. In addition, incorporating RSFMA increases the stiffness (elastic modulus, storage modulus) of composite hydrogels, which is more dominant in promoting the orderly alignment and extensive diffusion of hDPSCs [267]. It has been demonstrated that high-stiffness scaffolds promote the differentiation of hDPSCs to form mineralized tissues. On the other hand, low-stiffness scaffolds promote the differentiation of hDPSCs to form soft pulp tissue [268]. Therefore, RSFMA/MeHA hydrogels that regulate stiffness are more conducive to controlling hDPSCs differentiation and new tissue formation. Previous studies have reported the beneficial effects of Tideglusib (Td) and melatonin (Mel) on pulp regeneration. Td promotes the healing process of hard tissues, such as bone or teeth [269,270,271], and Mel promotes odontogenic differentiation [272,273]. Based on this, Atila et al. combined an injectable GelMA/PecTH hydrogel with core/shell PMMA/SF fibers loaded with Mel in the core and Td in the shell, which were successively released to promote hDPSCs proliferation and odontogenic differentiation [274].

## 8. Application of SF Hydrogel in the Tympanic Membrane

The tympanic membrane (TM) is an elastic, grayish, translucent film between the external and middle ears. It has an epithelial layer (outer layer), a fibrous layer (middle layer), and a mucosal layer (inner layer) and consists of keratinocytes, fibroblasts, and collagen (types II and III). Any mechanical injury, infection, and changes in pressure can lead to perforation of the tympanic membrane, causing hypoacusis or even loss of hearing [275,276,277]. Tympanoplasty is a common procedure to treat tympanic membrane perforations, and there is evidence that hearing may improve after surgery but may not be fully recovered [278,279,280]. Although various graft materials have been tried for tympanic membrane repair, the results have been unsatisfactory [281]. With the emergence and development of tissue engineering, researchers have explored the application value of various biomaterials, including fibroin, in artificial eardrum engineering [282]. SF hydrogels have not been reported in this field, but membranes and scaffolds made from SF have progressed (Table 4).

Levin et al. demonstrated that SF scaffolds successfully supported the growth and adhesion of human tympanic membrane keratinocytes while maintaining their cell lines [283], which is consistent with other findings [284]. Regenerated SF membranes (RSF) have good sound transmission ability and tensile strength to cartilage [285]. Shen et al. demonstrated that SF scaffolds (SFSs) improved the structure and function of TM restorations compared with conventional paper patches, resulting in complete closure of the perforation site and early restoration of hearing using a guinea pig acute TM perforation model [286]. At the same time, SFSs are well tolerated by the host and produce therapeutically acceptable tissue responses, including minimal inflammation, marked neovascularization, and scaffold degradation [287]. Lee et al. prepared electrospun polycaprolactone (PCL)/SF composites and combined them with human cord serum (UCS), which showed significantly improved bioactivity, rapid healing of tympanic membrane perforations, and enhanced hearing recovery compared with pure SF [288]. A cohort study of an SF patch to treat chronic tympanic membrane perforation showed that compared with traditional tympanoplasty, the SF patch technique had the advantages of a shorter operation time, easy operation, no need for skin incision during the operation, no need for dressing after the operation, and reduced incidence of complications such as otorrhea and improve patient satisfaction [289].

**Table 4 gels-09-00431-t004:** A summary of different types of silk fibroin hydrogels used in cornea, dental and Tympanic Membrane.

	Material(s)	Experimental Mode	Synthesis Method	Function	Ref.
Cornea	SF, PVA, nano-Hydroxyapatite, Dimethyl sulfoxide (DMSO), Genipin (GP)	In vitro	Chemical cross-linking agents	HCFs proliferate in composite hydrogel pores	[221]
	SF, PA, N,N′-MBAAm(cross-linker)	In vitro	Chemical cross-linking agents	Promote keratinocyte migration	[223]
	Acetone Optima, SF solution	In vitro	Chemical cross-linking agents	Provide a place for HCECs to adhere and proliferate	[225]
	SF solution, GelMA, Irgacure 2959(photoinitiator)	In vitro	Photocrosslinking	SF improved the mechanical stability and adhesion of the composite hydrogel and reduced the degradation rate	[14]
	Glycidyl methacrylate, SF solution	In vitro	Photocrosslinking	Effectively inhibit bacterial growth by loading antibiotics	[236]
	SF, flavin mononucleotide (FMN)	In vitro	Photocrosslinking	Change the optical thickness of the cornea	[241]
Dental	RSF, carbamide	In vitro	Shear force	Support the proliferation of hDPSCs	[263]
	SF, Riboflavin (photoinitiator)	In vitro	Photocrosslinking	Support the proliferation of hDPSCs	[264]
	SF	In vitro	Self-assembly	Supporting stem cell proliferation of human deciduous teeth	[265]
	SF, HA, Irgacure 2959(photoinitiator)	In vitro	Photocrosslinking	Control the differentiation of hDPSCs and promote the formation of new tissues	[266]
	SF, Td drug, Mel crystalline, Type A porcine skin gelatin, Irgacure 2959, methacrylic anhydride, High methoxyl Pectin, Thiogylcolic acid	In vitro	Electric field	Promote hDPSCs proliferation and odontogenic differentiation	[274]
Tympanic Membrane	SF scaffolds (SFS)	In vitro		Complete closure of the piercing site and early recovery of hearing	[286]
	Polycaprolactone (PCL), SF, human umbilical cord serum (hUCS)	In vitro		Effectively heals perforated tympanic membranes and improves hearing	[288]

## 9. Conclusions

In conclusion, SF hydrogels are widely used in human tissues. In the cartilage section, we introduced the main uses of SF hydrogels, namely, embedding drugs to treat chondritis and adding biomaterials to accommodate chondrocyte regeneration. In manufacturing hydrogels, in addition to considering the advantages and disadvantages of cross-linking methods, 3D bioprinting can be considered because its fabrication process is more flexible and accurate than freeze-drying or self-assembly. It has great potential in constructing TE cartilage biomimetic structures, although it has the shortcomings of using toxic organic solvents and insufficient mechanical strength at this stage. SF gels can play a role in bone tissue repair and regeneration by combining autologous growth factors (BMP-2 and VGEF) and inorganic minerals (HAP and LAP). Currently, SF gels are often used for skin tissue engineering by polymer blending or surface modification with commonly used additives. In addition to the traditional use of combining anti-inflammatory biomaterials such as plant extracts, the mechanical properties of SF gels are well-suited for emerging flexible electronic devices, which is a cause for concern. Although some studies have shown that SF gels can be applied to TE cornea and vision correction, dental pulp regeneration, and artificial tympanic membrane, few relevant studies are available, and further studies are still required.

## 10. Discussion

The commonly used SF gel additives include hyaluronic acid (HA), chondroitin sulfate (CS), chitosan (CTS), hydroxyapatite, polyethylene glycol, Laponite, etc.

HA, a biopolymer, was first discovered in bovine vitreous and later found in the synovial fluid, the umbilical cord, and the combs of chickens [290]. It has water solubility, rheology, and high viscosity, and HA of different molecular weights can promote or inhibit inflammation and angiogenesis [291]. It is mainly used as an injection for the conservative treatment of osteoarthritis and tendon disease and as a viscosupplement in joint lubrication [290]. It can also be used in ophthalmology as a substitute for vitreous humor, as a lubricating eye drop for dry eye syndrome, to improve the hydration performance of contact lenses, and for drug delivery to the eye [292].

CS, a linear polysaccharide, is mainly extracted from animal cartilage and plays a role in the core protein scaffold [293]. On the one hand, it can serve as a cell surface receptor for pathogens such as malaria parasites and herpes simplex virus, which infect hosts by attaching to CS chains [294]. Moreover, different structures of CS can act as extracellular signaling molecules, participating in intramembranous ossification to promote bone formation or influencing the growth of neural processes in the central nervous system. It also participates in the pulmonary metastasis of Lewis lung carcinoma cell lines and drugs. Additionally, CS affects the plasticity of neurons in the brain development process and the maturation of embryonic skeletal muscles [293]. On the other hand, CS has a water retention effect; is anti-inflammatory, antioxidant, anti-tumor metastasis, and proliferation; is an anticoagulant and anti-thrombotic agent; regulates immunity; and can transport and release drugs [295]. Therefore, it is used in treating cardiovascular and CNS diseases and as a nutritional supplement, and it is an over-the-counter drug for treating osteoarthritis. CS can act as or modify drug (or protein) carriers for enhancing efficacy and targeting effects [296]. Furthermore, it can encapsulate nucleic-acid-related polymers to protect them from degradation in the cytoplasm due to its anionic effect [297]. Additionally, it can be used as an adjuvant for vaccines to enhance the immunogenic response [298].

Due to active functional groups, CTS can easily be modified chemically to enhance its physicochemical properties to serve as a drug carrier and as a sustained release system, gene therapy vector, drug material, and mucosal immunomodulator. In addition to its application in synthesizing composite materials for bone and dental purposes, CTS can be used as an antibacterial agent or coating to produce antibacterial plastics, metals, fibers, ceramics, and other materials in various forms [299,300]. In addition to gels, CTS can also serve as a drug carrier in the form of microspheres, nanoparticles, micelles, etc. [301] and can be used for mucosal vaccination to enhance the therapeutic effect. Moreover, CTS can be used as a fluorescent probe, providing high selectivity and sensitivity and better cell compatibility, which has enormous potential in cell imaging and heavy metal ion recognition [302].

HAP is a type of bioactive calcium phosphate. In addition to its application in bone tissue engineering, it is also widely used in commercial oral care products, such as toothpaste and mouthwash, and for enamel and dentin remineralization, reduction of tooth sensitivity, prevention of caries and biofilm formation, and tooth whitening. As an anti-cariogenic agent, it is superior or equivalent to fluoride-containing toothpaste [303,304]. In addition, HAP nanoparticles can be used for cancer treatment, magnetic imaging, and targeted killing of cancer cells [305].

Biological materials such as agarose, lignin, chitosan, hyaluronic acid (HA), alginate, gelatin, GelMA, and polylactide (PLA) are commonly used for hydrogel synthesis in tissue engineering (TE). The agarose-based gel has mechanical properties similar to natural cartilage but is non-degradable and immunogenic, with limited processing ability [44]. In contrast, SF has better biological performance for in vivo injection or implantation. Lignin, a biopolymer widely present in plants, is made into gel due to its biocompatibility, biodegradability, and low toxicity. However, its preparation relies on an interpenetrating polymer network, while SF gel preparation and cross-linking are relatively simple and diverse. In addition, the lignin-based gel has more rigidity. In contrast, SF gel has better flexibility and biodegradability, making it more versatile in tissue regeneration, such as through injection filling of cartilage defects or flexible skin sensors [306]. Although chitosan has been mentioned several times as a synthetic additive for SF gel, CTS itself is often used as a scaffold material and can be converted into a gel, a powder, a film, microspheres, sponges, and other forms. The CTS-based gel is often applied for drug delivery, such as targeted, transdermal, and oral delivery. However, it is less tough than SF and may cause allergic reactions, while SF is relatively more suitable for tissue reconstruction [307]. Similarly, hyaluronic acid, alginate, and gelatin are also commonly used natural polymers for gels. However, HA has poor mechanical properties, gelatin may cause allergic reactions, and sodium alginate has poor stability under physiological conditions, low tissue adhesion, and poor mechanical properties [5,307]. GelMA, made from gelatin and methacrylic acid, has properties similar to a natural ECM and is a traditional hydrogel synthesis material. It has good biocompatibility, processability, and tunable mechanical properties and is easily cross-linked by light, making it widely applicable in a hybrid gel. However, GelMA itself does not have the stability of SF; therefore, SF has more diverse bonding modes, and more hybrid processing forms are yet to be studied [308]. PLA is also commonly used as a drug carrier or scaffold. In contrast to lignin, the mechanical properties of PLA water gels are usually soft, making it difficult to use them in medical implants. In addition, the cross-linking of PLA gel is mostly pH- or temperature-sensitive self-assembly, whose cross-linking methods are less diverse than SF-based gels [309]. Furthermore, as a synthetic polymer, it may elicit immunogenicity, which is a common drawback of synthetic polymers such as polyurethane (PU), poly(lactide-co-glycolide) (PLGA), and polycaprolactones (PCL) [5,309].

## Figures and Tables

**Figure 1 gels-09-00431-f001:**
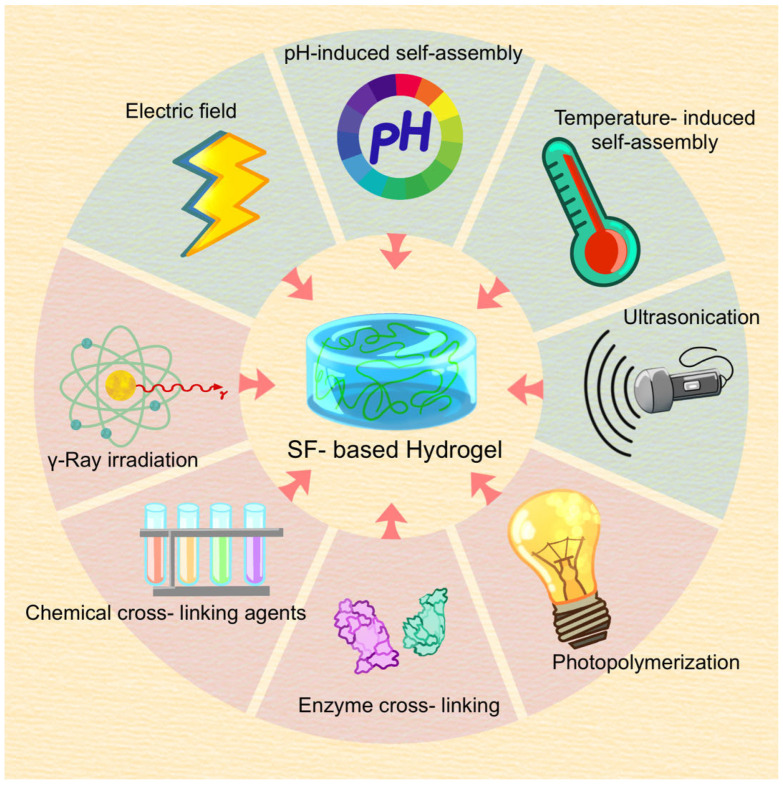
SF hydrogel crosslinking methods include physical and chemical crosslinking. Methods used for physical crosslinking include temperature and ph-mediated self-assembly, ultrasonication, electric fields, etc. Methods used for chemical crosslinking include chemical cross-linking agents, photopolymerization, γ-Ray Irradiation, enzyme cross-linking, etc.

**Figure 2 gels-09-00431-f002:**
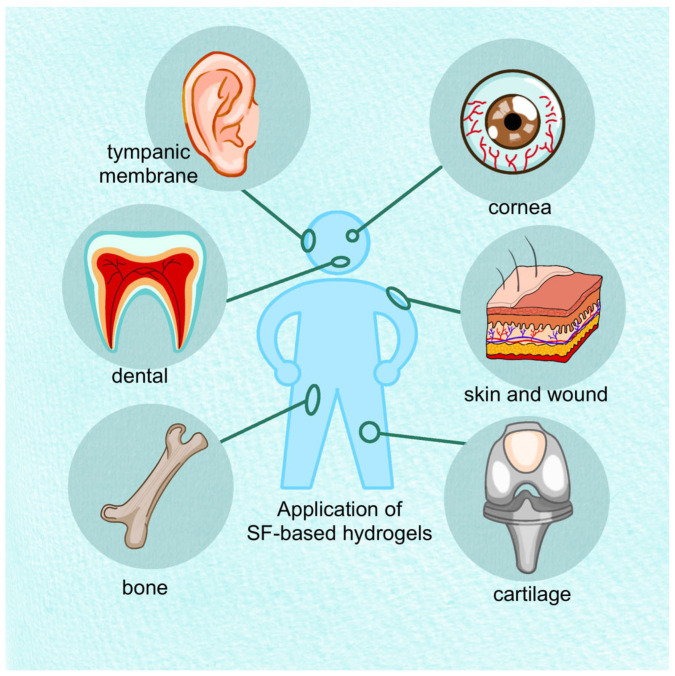
In human tissues, SF-based hydrogels can be applied to bone, cartilage, skin and wound, cornea, teeth and tympanic membrane.

## Data Availability

No new data were created.

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
