# Peer review of "Application of Silk-Fibroin-Based Hydrogels in Tissue Engineering"

_gels, 2023, doi:10.3390/gels9050431_

Round 1
Reviewer 1 Report
This review article discusses the potential applications of silk fibroin-based hydrogels in tissue engineering, including cartilage, bone, skin wounds, cornea, tympanic membrane, and teeth. Globally this is a well-structured and scientifically sound manuscript and could positively contribute wound healing field.
Below I point out some comments and issues that the author should consider for this manuscript:
1. While the abstract already provides relevant information, it could be enhanced by including a comprehensive overview of the review's structure.
2. The introduction provides a good overview of the current challenges faced in treating organ defects and highlights the potential of silk fibroin-based hydrogels in tissue engineering due to their excellent properties. However, while it is mentioned that the paper "systematically reviews the current manufacturing methods of SF hydrogels and the relevant applications of various SF hydrogels," it could be beneficial to clarify the specific goals or objectives of the review. In addition, the introduction includes a lot of background information on the properties and structure of silk fibroin. While this information is essential, it could be condensed to focus more on the critical challenges of current methods for treating organ defects and the potential of SF-based hydrogels to address these challenges.
3. Section 2 offers a good explanation of the preparation of SF gels, but making some small adjustments could make it even clearer and easier to understand for readers.
a. In the first paragraph, the sentence "SF and silk fibroin protein together form silk" is redundant, as SF is short for silk fibroin. It would be clearer to say "The production of SF gels requires obtaining silk fibroin raw materials, which are typically obtained by removing the sericin protein through a process called degumming."
b. In the second paragraph, the phrase "Currently, many silk protein-based hydrogel products are available" is not necessary and could be removed without affecting the meaning of the paragraph.
c. In the third paragraph, the sentence "Various physical methods, such as hydrogen bonding, hydrophobic interactions, electrostatic interactions, and ion interactions, are used to induce silk fibroin molecules to adopt a lower β-sheet conformation, which then binds together via non-covalent bonds to form a hydrogel" is quite long and complex. It might be clearer to break it up into two or three sentences for easier readability. Additionally, the phrase "lower β-sheet conformation" might not be clear to all readers, so it could be helpful to provide a brief explanation or definition.
d. In the fourth paragraph, the sentence "Ultrasonic power output, duration, and silk fibroin concentration are important factors that can influence the state of the hydrogel" is a bit vague. It might be clearer to explain what specific aspects of the hydrogel's state these factors can affect.
4. Section 3 comprehensively reviews various methods for producing silk-based hydrogels for cartilage tissue engineering. Providing a brief overview or background of cartilage tissue engineering and the current challenges and limitations might help contextualize the importance and relevance of silk-based hydrogels for cartilage tissue engineering. In addition, more direct comparisons between the different methods and their respective advantages and disadvantages help readers better understand which method might be most suitable for their specific application or research question. While the section provides a lot of information about the methods used and the outcomes observed, there is less interpretation or analysis of the results or their implications for future research or clinical applications.
5. Section 4 has some quite long and complex sentences, making them difficult to follow. Breaking them up into shorter sentences could make it easier for readers to understand.
6. Section 5 could be enhanced for its readability and clarity: Some of the sentences are long and complex, making them challenging to follow. while the section provides a wealth of information about the use of SF in various biomedical applications, it could be strengthened by discussing potential limitations or challenges to its use and ways that researchers are addressing these issues
7. Section 6 could also be improved if the author add more details about the limitations of the current bioengineered cornea techniques and how SF hydrogel could potentially address these limitations. Additionally, the author could provide more examples of how SF hydrogel has been tested in vivo and how it compares to other bioengineered cornea materials in terms of biocompatibility and effectiveness.
8. Section 9. discussion the D should be capitalized. Provide references or citations for any claims or data presented in the section. With some restructuring and editing, the discussion section can be clearer and more accessible to readers.
9. Conclusions separate section could be beneficial.
Some of the sentences are long and complex, making them challenging to follow.
Reviewer 2 Report
Authors have conducted a review on the applications of silk fibroin-based hydrogels in tissue engineering.
A proper title for a systematic review must include the phrase “A systematic review.” at the end of it. The phrase “systematic review” is used for the first time at the end of the first paragraph of the introduction.
In the list of authors at the beginning of the review, each author has a different number (supra-scripted) indicating that each author has its own unique affiliation. However, there is only one affiliation detailed. If all 4 authors have the same affiliation, then they all should refer to the same number (i.e., 1) next to their name. And if they do have different affiliations, then the remaining 3 affiliations must be detailed as well.
Through my extensive search I found an oddly similar review article titled “Silk Fibroin-Based Biomaterials for Tissue Engineering Applications” which was published in MDPI’s Molecules journal on April 25th 2022. I find the title, main purpose, results and conclusion of these 2 reviews very similar. Can the authors of this newly submitted review justify the novelty of their paper? Was there really a need for another review on the exact same subject less than a year apart?
I personally believe – from what I have read and reviewed on these 2 review papers – that this review simply lacks novelty and fresh data, hence, does not hold much value.
Abstract:
The third line of the abstract is very confusing. “Traditional scaffold materials suffer from poor mechanical properties, large differences, and high risk”.
What exactly is the “large difference” referring to? Large differences in size? physical and chemical properties? biochemical features?
Same issue with “high risk”: high risk of what exactly?
Authors need to use strict, short and clear sentences in their abstract. Otherwise, readers would be left with confusing and immature sentences as such mentioned above.
The word “various” has been overused in the abstract. I suggest assessing similar words or even give an example or two of the “various” mentioned objects/subjects.
A proper abstract for a review paper must be concise and comprehensive. It must be short while highlighting the key findings of the review. This abstract has given a perfunctory, negligent and rushed overview of the assessment of silk fibroin in general. There is not a single sentence in this abstract that brings any new valid or useful information to the average readers’ knowledge.
“Moreover, SF is water-soluble, low-cost, and easy to use, making it a versatile material that is widely applied in various biological fields, particularly in tissue engineering.”. The SF is “low-cost and easy to use”, compared to what? If this statement is compared to other tissue-engineering materials/scaffolds, then at least a couple of them must be mentioned. Otherwise, this statement is baseless.
I do understand that the word limit on the abstract does not allow authors to fully explain the rationale, importance and necessity of their review. However, authors must not fill the already short-worded abstract with useless data. They should take advantage of the abstract’s word limit and only highlight data that is truly crucial in raising readers’ excitement/interest in reading the whole manuscript.
This abstract needs to be completely re-written with a special attention to details that truly matter, instead of putting vague and general sentences next to each other, only for the sake of meeting the abstract’s word limit.
Keywords:
The word “application” is not a proper keyword in any context. I suggest removing “application” from the keywords and adding “tissue engineering” and “scaffolds” instead.
Introduction:
The reference number “13”, does not have a proper citation: is it a journal, book, conference or a webinar? year of publication? Pages? author list?
The last sentence in the first paragraph of the introduction (“SF-based hydrogels have gained extensive attention in the fields of artificial and regenerative tissues due to their excellent properties”) does not have any reference. Given that authors are suggesting a well -established fact and not just their opinion, they absolutely need to refer to a newly-published study that has reported these statements.
The first paragraph is suggesting that the commonly assessed methods of treating organ defects with tissue engineering, have major issues such as donor shortages, rejection, inflammatory reactions, etc. First of all, if these statements are backed by any sources, authors must include some examples and clearly detail their challenges. Additionally, to my knowledge, there are numerous tissue engineering scaffolds/methods that do not require tissue grafts (e.g., autologous or allogenic transplantations) and have resulted in remarkable outcomes only by applying/seeding stem cells into scaffolds/hydrogels.
Authors need to execute a better background search on the most current technologies and methods assessed in tissue engineering for small and large human defects.
The introduction as it is right now, does not hold much value in regards to their judgement on the superiority of SF to other tissue engineering biomaterials. Given the fact that tissue engineering has been a well-established field for many in vitro, in vivo, and clinical human trials and experiments in past decades, I believe that authors can do a much better job in rationalizing the importance of assessing SF. And with proper referring to previous works in this field, readers would have a better appreciation of their findings and judgements.
This introduction is not acceptable for a proper systematic review.
Methods, Materials, and results:
A proper systematic review must include the following:
1. A comprehensive introduction on the background of the subject, the previous and current status of the leading technologies in this field, and the reason that this systematic review is not only rationalized but necessary.
2. Methods and materials: authors need to clearly state that their review has followed all of the PRISMA 2020 guidelines. They need to prepare a PRISMA flow diagram of their included studies, the screening and selection process. Authors are highly suggested to register their review to PROSPERO before submission to any journal and provide their PROSPERO registration ID in their manuscript. They need to clearly state that each phase of the review was performed by whom. They have to provide a table for their search queries indicating that which search terms were assessed for each data base and how many studies were initially found. They have to clearly state the purpose of their study in the PICO format.
3. Results must be comprehensive and detailed. Every subtitle must be chosen carefully. Tables and figures must all have a short description.
4. And eventually an appropriately accurate discussion on their extracted results, and their personal judgments compared to the previously published studies.
From what I what have learned, the assessment of fibroin-based hydrogels and materials are widely diversified. I suggest authors first of all correct their style of reporting in a proper systematic review, and focus solely on a specific field of fibroin-based material assessments instead of a rushed and undetailed review on all of the categories of assessments.
Tables:
None of the tables in this review have a proper numbering system or description. Each table must have its number and description attached above the table, and a list of abbreviation/explanations on the bottom of the table.
In general, this review is particularly weak both in the English-language grammar and choosing proper English words. I highly suggest revisiting this manuscript and assessing well established English-language tools to help with their writing. I do believe that authors need to seriously reconsider if their paper is a systematic review or just a superficial and rushed review of literature. Following PRISMA guidelines and a PROSPERO registration ID are a must for any systematic review. Tables have a proper way of execution that must not be ignored. Any statement that is not authors’ opinion must have a proper reference.
From my extensive search on multiple data bases, I found out that the assessment of silk-fibroin-based materials/hydrogels/scaffolds in tissue engineering is widely versifies. Hence, a proper systematic review that follows all of the well-established guidelines can be extremely valuable and useful. Unfortunately this submitted review paper does not fulfill the requirements of a proper systematic review.
The only exception that would make this review acceptable would be a direct invitation from Gels. Otherwise, this style of reporting and reviewing is far from appropriate.
Overall, I do believe that this subject does have potentials for the future of tissue engineering. However, this review is poorly written, the abstract needs to be completely re-written, does not follow the PRISMA , Cochrane or any other guideline, is not registered in PROSPERO and needs serious revisions in choosing proper English words and respecting the English-language grammar. I suggest rejecting this review with no revisions accepted.
In general, this review is particularly weak both in the English-language grammar and choosing proper English words. I highly suggest revisiting this manuscript and assessing well-established English-language tools to help with their writing.
The third line of the abstract is very confusing. “Traditional scaffold materials suffer from poor mechanical properties, large differences, and high risk”.
What exactly is the “large difference” referring to? Large differences in size? physical and chemical properties? biochemical features?
Same issue with “high risk”: high risk of what exactly?
Authors need to use strict, short and clear sentences in their abstract. Otherwise, readers would be left with confusing and immature sentences as such mentioned above.
The word “various” has been overused in the abstract. I suggest assessing similar words or even give an example or two of the “various” mentioned objects/subjects.
Round 2
Reviewer 2 Report
The authors have done a great job of complying with the suggested changes made by me. They have also clarified that they do not tend to present their paper as a systematic review and that their paper is only a review of literature. Their abstract and introduction were filled with vague, superficial, and undetailed statements that made their manuscript extremely unreliable. However, their revised manuscript has significantly improved. I suggest accepting this paper.